# Crosstalk between chromatin state and ATM signalling in DNA damage-induced transcription stress

Irene Salas-Armenteros � , Maarten Klunder, Wim Vermeulen ✉ & Maria Tresini ✉

## Abstract

The DNA Damage Response (DDR) is a highly regulated process that safeguards genomic integrity against DNA lesions. Increasing evidence supports a reciprocal relationship between damaged chromatin architecture and the signalling pathways that coordinate the DDR. However, the mechanisms underlying this interplay in response to transcription-blocking DNA lesions remain largely unexplored. Here, we show that stalling of RNA polymerase II (RNAPII) at such lesions induces local chromatin acetylation, mediated primarily by the histone acetyltransferase p300. The resulting chromatin relaxation stimulates the dissociation of mature co-transcriptional spliceosomes from nascent RNA and promotes RNA:DNA hybrid (R-loop) formation, leading to ATM activation. In turn, activated ATM modulates chromatin conformation by phosphorylating histone H2A.X and triggering p38MAPK/MSK1-dependent histone H3S10 phosphorylation. Our findings highlight the cross-regulation between chromatin state and ATM signalling as a key component of the cellular response to transcription stress.

**Keywords** DNA Damage Response Signalling; Transcription Stress; Chromatin acetylation; Spliceosome; R-loops
**Subject Categories** Chromatin, Transcription & Genomics; DNA Replication, Recombination & Repair

## Introduction

DNA damage-induced genomic instability is a fundamental contributor to aging and various human pathologies including cancer and neurodegeneration (Barzilai et al, 2017; Negrini et al, 2010; Niedernhofer et al, 2018; Schumacher et al, 2021). Cells have evolved intricate DNA Damage Response (DDR) mechanisms to preserve genomic integrity and tissue homeostasis. Upon DNA damage, the DDR coordinates the recruitment, activation, and orchestration of specialized protein networks to halt cell cycle progression, repair DNA, reset transcriptional programs, or, as a last resort, trigger senescence or apoptosis.

Efficient execution of these programs requires extensive chromatin reorganization, primarily achieved by nucleosome remodellers and chromatin modifiers that control chromatin compaction through post-translational modifications (PTMs) of core histones (Hauer and Gasser, 2017; Jeggo et al, 2017; Karakaidos et al, 2020; Kim et al, 2019; Ortega et al, 2021). In parallel, DDR signalling influences chromatin architecture both near damaged DNA—facilitating repair—(Becker et al, 2014; Berkovich et al, 2007) and at distal sites to regulate transcription programs critical for checkpoint control and damage recovery (Goodarzi et al, 2011; Iannelli et al, 2017; Kruhlak et al, 2006; Li and Cucinotta, 2020; Shanbhag et al, 2010; Ziv et al, 2006). The crosstalk between chromatin structure and DDR signalling in response to certain types of lesions, as for example double-stranded DNA breaks (DSBs), has received considerable attention over the years (Goodarzi et al, 2011; Iannelli et al, 2017; Kruhlak et al, 2006; Li and Cucinotta, 2020; Shanbhag et al, 2010; Ziv et al, 2006). However, the mechanisms underlying this interplay in response to transcription-blocking DNA lesions (TBLs) remain largely unexplored.

DNA helix-distorting lesions within the template strand of transcribed genes inhibit transcription by obstructing the forward translocation of RNA polymerase II (RNAPII) (Brueckner et al, 2007). In higher eukaryotes, RNAPII transcription is coupled to nascent RNA splicing and is subject to extensive feedback control dictated by the chromatin (Bentley, 2014; Imbriano and Belluti, 2022; Luco et al, 2011; Luco et al, 2010; Rahhal and Seto, 2019; Skalska et al, 2017; Tresini et al, 2016). RNAPII stalling at transcription-blocking lesions, stimulates the active dissociation of mature co-transcriptional spliceosomes from stalled elongation complexes (Tresini et al, 2015). In the absence of steric hindrance imposed by the spliceosome, unprocessed, nascent RNA can readily hybridize with the template strand within the negatively super-coiled DNA helix upstream of the elongation complex. This leads to formation of aberrant and potentially pathogenic nucleic acid structures (R-loops), consisting of an RNA:DNA hybrid across from ssDNA (Garcia-Muse and Aguilera, 2019; Li and Manley, 2005; Petermann et al, 2022; Salas-Armenteros et al, 2017; Stolz et al, 2019; Tresini et al, 2015).

Department of Molecular Genetics, Erasmus MC Cancer Institute, Erasmus University Medical Centre, Rotterdam 3015 GD, The Netherlands.
✉ E-mail: w.vermeulen@erasmusmc.nl; m.tresini@erasmusmc.nl

R-loop formation precedes and contributes to ATM activation, through mechanisms that vary with the cell cycle phase. In non-replicating cells, ATM activation relies exclusively on R-loops (Tresini et al, 2015) whereas in S-phase cells, it can also be triggered by DSBs arising from NER processing of R-loops (Sollier et al, 2014; Tresini et al, 2016) or replication fork stalling (Crossley et al, 2019). Following activation, ATM phosphorylates its canonical targets and independently influences spliceosome organization (Tresini et al, 2015). The combined functions of ATM drive extensive transcriptional reprogramming, as evidenced by over 4500 ATM-dependent splicing and gene expression changes in non-replicating cells exposed to UV radiation (Tresini et al, 2015).

During transcription elongation, RNAPII operates in a highly dynamic chromatin environment where chromatin relaxes in front of the polymerase and subsequently re-assembles upstream of the elongation complex. This transcription-associated chromatin reorganization is a tightly regulated process, reciprocally linked to RNAPII processivity and associated co-transcriptional processes, including pre-mRNA splicing (Agirre et al, 2021; de Almeida and Carmo-Fonseca, 2014; Luco et al, 2010) and R-loop formation (Bayona-Feliu et al, 2021; Sanz et al, 2016). Whether and how RNAPII stalling at TBLs influences chromatin dynamics and downstream damage signaling remains largely unknown. Here, we show that RNAPII arrest at these types of lesions initiates a chromatin-modifying cascade, beginning with local histone acetylation primarily mediated by the histone acetyltransferase p300. This acetylation, typically associated with relaxed chromatin, facilitates spliceosome displacement, promotes aberrant RNA:DNA hybrid formation, and activates ATM signaling. In turn, activated ATM contributes to chromatin remodeling through histone phosphorylation, establishing a feedback loop that reinforces the transcriptional stress response (Berger et al, 2017; Li and Cucinotta, 2020). Together, our study reveals that chromatin modulation is not just a consequence of transcription stress, but a critical upstream and downstream regulator of the damage response to transcription blocking DNA damage.

## Results

### Spliceosome eviction from hyperacetylated chromatin

Chromatin modifications are well established as key regulators of the cellular response to replication stress, double-strand breaks, and DNA helix-distorting lesions in non-transcribed regions, where chromatin decompaction is thought to facilitate access by repair systems (Dabin et al, 2023). In contrast, their role in the response to transcription-blocking lesions (TBLs) remains underexplored. This gap is likely due to the assumption that chromatin decompaction is unnecessary for TBL response, given that transcription typically occurs in already relaxed chromatin. Our studies aim to address this gap by examining changes in chromatin modifications near TBL-stalled RNAPII and their impact on the downstream activation of DDR signalling.

In preliminary studies, we identified a range of UVC-induced changes in post-translational modifications (PTMs) of core histone H3 (Figs. EV1A and 1A,B). These included increased lysine acetylation, a phenomenon previously linked to chromatin relaxation and enhanced DNA damage repair in replicating cells (Brand et al, 2001; Guo et al, 2011; Li et al, 2022; Sustackova et al, 2012).

To investigate whether the typically recompacted (hypo-acetylated) chromatin upstream of transcription elongation complexes is similarly modified upon TBL-stalling of RNAPII, we leveraged the dynamics and binding position of Cockayne Syndrome Group B (CSB) relative to stalled RNAPII (Kokic et al, 2021; Llerena Schiffmacher et al, 2023). Upon UV irradiation, we observed sustained acetylation of immuno-precipitated CSB-associated mononucleosomes (Fig. EV1B) from RPE1 CSB–mScarlet-I KI cells (van Sluis et al, 2024) (Fig. 1C). Considering the reciprocal coupling between chromatin acetylation and elongating RNAPII progression, the near-complete transcription inhibition induced by the applied UV dose (20 J/cm$^2$, 1–2 h) (Fig. EV1C), should theoretically suppress histone H3 acetylation both upstream and downstream of TBL-stalled RNAPII. However, since CSB binds strongly to both RNAPII and DNA upstream of the stalled polymerase (Kokic et al, 2021), we inferred that histone H3 in CSB-associated mononucleosomes might be re-acetylated to relax chromatin upstream of TBL-stalled RNAPII. Building on our previous findings that TBL-induced ATM activation depends on spliceosome eviction from transcription elongation complexes and subsequent R-loop formation (Tresini et al, 2015), we next asked whether TBL-induced chromatin acetylation plays a regulatory role in this process.

All experiments were conducted in TERT-immortalized diploid human dermal fibroblast (HDF) cell lines, C5Ro and VH10, which were synchronized in quiescence. These cell lines and conditions were specifically selected to capture the DNA damage response (DDR) to transcription-associated stress, while excluding replication-associated signalling that could otherwise confound data interpretation (Marti et al, 2006). Experiments were performed following pharmacological or genetic inhibition of histone modifiers prior to the induction of TBLs.

We previously demonstrated, using a combination of biochemical and imaging approaches, that TBLs specifically trigger the dissociation of proteins and snRNAs associated with late-stage spliceosomes from nascent RNA (Tresini et al, 2015). This DNA damage-induced spliceosome displacement occurs specifically in response to TBLs generated by UV irradiation or Illudin S treatment, but not in response to other types of DNA damage that do not affect RNAPII processivity (Tresini et al, 2015). Our earlier studies also established the high sensitivity and reproduci-bility of splicing factor (SF) fluorescence recovery after photo-bleaching (FRAP) as a quantitative method to monitor spliceosome remodelling in real-time (Tresini et al, 2015). Therefore, we employed FRAP to evaluate the impact of pharmacologically induced structural chromatin changes on SF mobility. The efficacy of the epigenetic drugs used was evaluated by their ability to modulate levels of histone H3 PTMs (Fig. EV1D,E). Core SFs of the U1 (U1a), U2 (SF3a1), U4 (PRPF3) and U5 (SNRNP40) snRNP complexes were selected to represent distinct stages of the splicing cycle (Will and Luhrmann, 2011) (Fig. 1D) and their dynamics (binding/dissociation) were measured by FRAP in living HDFs engineered to stably express GFP-tagged SFs (Fig. EV2A,B). Histone hyperacetylation induced by UV-irradiation (Figs. 1A,B and EV1A) was paralleled by the selective chromatin release of SFs participating in mature spliceosomes (U2 and U5 snRNPS) and not those associated with earlier stages of the splicing cycle (U1 and U4 snRNPs) as evidenced by an increase in their FRAP-measured

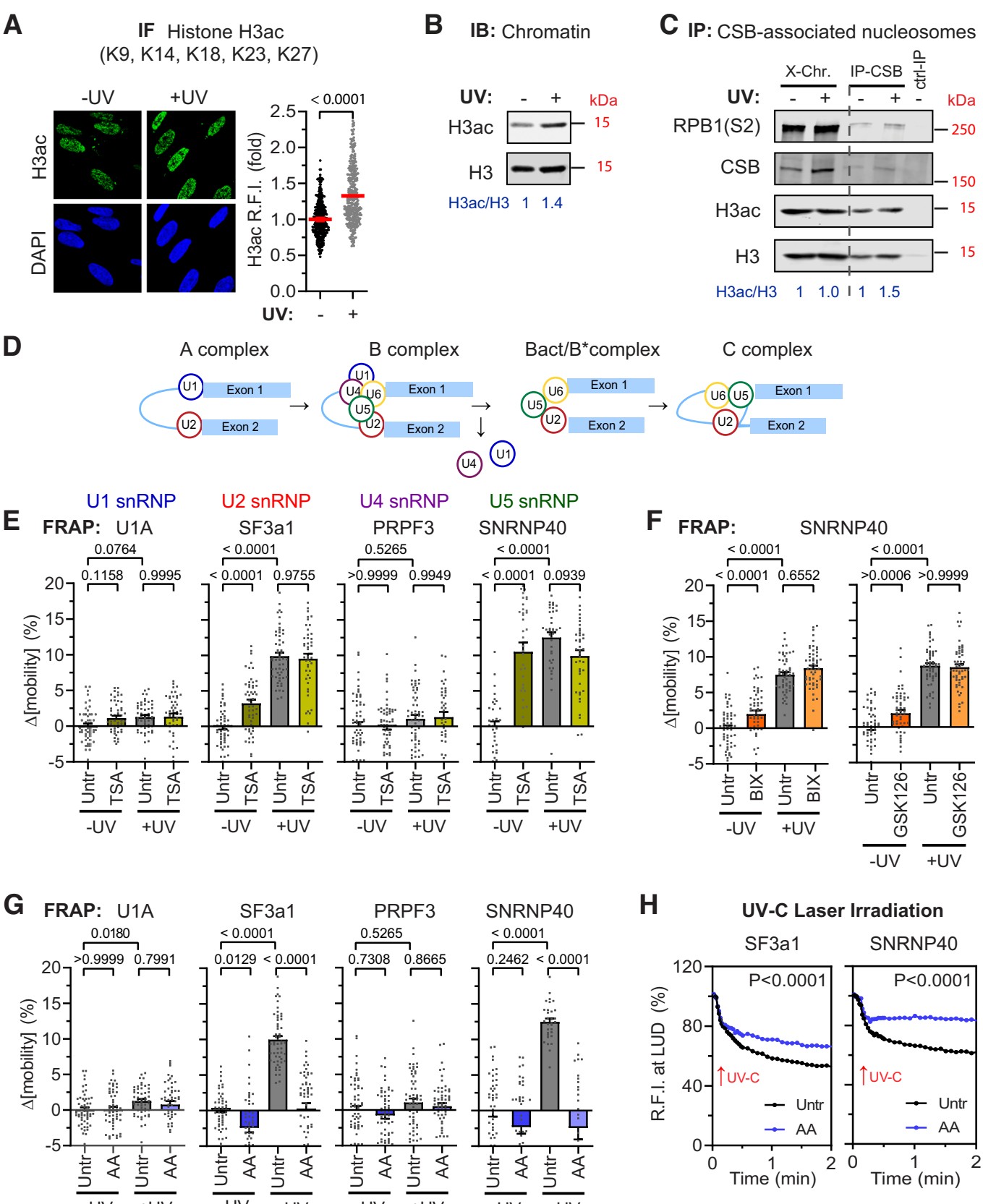

**Figure 1. Displacement of mature spliceosomes from UV-damaged chromatin is regulated by chromatin modifications.**

(A, B) Histone H3 acetylation (H3ac) in unperturbed and UV-irradiated (20 J/m², 30 min) quiescent human dermal fibroblasts (HDFs). (A) Representative immunofluorescence images and quantification of H3ac. Cells were detergent-extracted prior to fixation; nuclei were stained with DAPI. Scale bar: 20 μm. Plotted are relative fluorescence intensities in irradiated versus non-irradiated cells. (B) Immunoblots showing acetylated and total histone H3 levels. (C) H3ac in CSB co-immunoprecipitated mononucleosomes. Cross-linked chromatin from non-irradiated and UV-irradiated (20 J/m², 2 h) RPE1 CSB–mScarlet-I knock-in cells was fragmented to mononucleosomes by sonication and nuclease digestion. Immunoblots detect CSB, elongating RNAPII (Ser2-phosphorylated RPB1), acetylated and total histone H3. (D) Schematic of spliceosome maturation. During splicing, splice site recognition by U1 and U2 snRNPs triggers recruitment of a preformed U4/U5/U6 tri-snRNP. Mature spliceosomes composed of U2, U5 and U6 snRNPs are formed following U1 and U4 dissociation and conformational rearrangements. (E, F) Chromatin relaxation promotes mobilization of mature spliceosomes. Mobility of indicated splicing factors (SFs), representing distinct snRNP complexes, was assayed by FRAP in unperturbed and UV-irradiated quiescent HDFs (20 J/m², 30 min). Chromatin relaxation was induced by (E) histone hyperacetylation via HDAC inhibition (Trichostatin A, TSA; 1 μM, 3 h) or (F) inhibition of repressive histone H3 methylation via methyltransferase inhibitors BIX01294 (BIX; 10 μM, 2 h) or GSK126 (10 μM, 2 h). (G) UV-induced SF mobilization is suppressed by chromatin hyperacetylation following HAT inhibition by Anacardic Acid (AA; 10 μM, 2 h). (E–G) Plotted are SF mobility changes (Δ[mobility] = mobility in treated minus untreated/non-irradiated cells) at 20 s post-photobleaching. (H) Suppressed SF displacement from compacted UV-C laser-irradiated chromatin. Displacement kinetics of SF3a1 and SNRNP40 from subnuclear UV-damaged regions. Cells were AA-treated (10 μM, 2 h) prior to irradiation. Plotted is fluorescence intensity at the irradiated site normalized to pre-irradiation level. Data information: (A) Mean ± SEM of $n = 347/322$ cells from three biological replicates; unpaired two-tailed $t$ test with Welch's correction. (E–G) Mean ± SEM of (left to right): (E) $n = 54/53/51/50$, $50/55/51/45$, $60/60/60/40$, and $34/37/39/42$; (F) $n = 54/54/54/54$, $54/54/54/54$; and (G) $n = 55/53/54/50$, $48/54/54/55$, $60/60/58/60$, and $45/41/34/27$ cells, all from three biological replicates. Brown–Forsythe and Welch's ANOVA with Games–Howell's multiple comparisons test (E, G) or one-way ANOVA with Bonferroni correction (F). (H) Mean ± SEM of $n = 37/35$ (SF3a1) and $n = 37/37$ (SNRNP40) cells from three biological replicates; repeated-measures two-way ANOVA. Source data are available online for this figure.

mobility [Δ[mobility]] (Figs. 1E and EV2C). Similarly, histone hyperacetylation by the broad range Histone Deacetylase (HDAC) inhibitor Trichostatin A (TSA) (Vanhaecke et al, 2004) (Fig. EV1D,E) induced selective chromatin release of mature spliceosomes (Figs. 1E and EV2C). TSA treatment prior to UV-irradiation had no further influence on SF mobility (Fig. 1E), indicating that mature spliceosome dissociation by TBLs is preceded by Histone acetylation.

To further investigate the role of chromatin relaxation on spliceosome dynamics, we abrogated methylation-related chromatin compaction using the histone methyltransferase (HMT) inhibitors, BIX01294 and GSK126. BIX01294 suppresses the G9a/GLP mediated deposition of H3K9Me2 and to a lesser extent of H3K9Me3 by G9a/GLP (Kubicek et al, 2007) while the EZH1/EZH2 inhibitor GSK126 suppresses H3K27me3 deposition (Pan et al, 2018) (Fig. EV1D,E). While typically associated with transcriptionally silenced chromatin (Hyun et al, 2017; Rice et al, 2003), Histone H3K9 and K27 methylations have also been observed in actively transcribed genes, playing a role in local chromatin compaction (Agirre et al, 2021; Allo et al, 2009; Estaras et al, 2013; Luco et al, 2011; Muniz et al, 2021; Saint-Andre et al, 2011; Schor et al, 2013; Vakoc et al, 2005; Zhou et al, 2014). Both HMT inhibitors exhibited similar, although milder effects on spliceosome mobilization as TSA (Fig. 1F) without affecting global Histone H3 acetylation levels (Fig. EV1E). The PTM-dependent SF mobilization by these three functionally distinct inhibitors, suggests a process that is likely driven by chromatin relaxation rather than by specific modifications of transcription-, or splicing-associated proteins (Cheng et al, 2007; Choudhary et al, 2009; Gunderson and Johnson, 2009; Gunderson et al, 2011; Schroder et al, 2013; Siam et al, 2019).

Having observed that Histone PTMs associated with relaxed chromatin, promote the mobilization of mature spliceosomes, we hypothesized that pharmacological inhibition of histone acetylation would suppress their dissociation from damaged chromatin (Tresini et al, 2015). Indeed, histone hypo-acetylation induced by the acetyltransferase inhibitor Anacardic Acid (AA) (Balasubramanyam et al, 2003) (Fig. EV1D,E), effectively suppressed the UV-induced mobilization of SFs associated with U2 and U5 snRNPs (Fig. 1G). Furthermore, when AA was administered prior to

UV-C microbeam irradiation, it suppressed the depletion of GFP-tagged U2 and U5 snRNP-SFs from irradiated sites, as monitored in living cells by confocal microscopy (Fig. 1H). Notably, global RNA synthesis remained unchanged following incubation with AA (or TSA) at the applied doses, indicating that the spliceosome's unresponsiveness to TBLs in the presence of AA is unlikely to result from a reduced rate of lesion recognition by RNAPII in highly compacted chromatin (Fig. EV2D). AA-induced loss of spliceosome's responsiveness to TBLs was also observed when cells were exposed to Illudin S, which generates specifically TBLs that are structurally distinct from UV-induced photolesions (Jaspers et al, 2002) (Fig. EV2E). Interestingly, the ability of AA to suppress spliceosome mobilization was restricted to TBL-generating treatments which preferentially target late-stage spliceosomes (Tresini et al, 2015). AA had no influence on spliceosome mobilization induced by the transcription inhibitor Flavopiridole (FLV), which prevents spliceosome assembly, or the U2 snRNP inhibitor Pladienolide B (PLB), which hinders spliceosome maturation (Kotake et al, 2007) (Fig. EV2E). We inferred that chromatin compaction suppresses predominantly the response of late-stage complexes to TBLs, rather than to stimuli targeting earlier stages of the splicing cycle.

## HATs promote spliceosome displacement from damaged chromatin

Having established that HAT inhibition (and ensuing chromatin compaction) blocks the spliceosome's response to TBLs, we used silencing strategies and chemical inhibitors to identify HATs that acetylate damaged chromatin at TBLs. Pharmacological inhibition of p300 and PCAF by CTK7A, or GCN5 by CPTH2, efficiently suppressed the UV-induced spliceosome mobilization (Fig. 2A) and displacement from UVC microbeam irradiated sites (Fig. 2B). It should be noted that both inhibitors were used at concentrations that suppressed histone H3 acetylation to similar levels and comparable to those of AA (Fig. EV1E). Combination of CTK7A and CPTH2 at suboptimal concentrations had a combined suppressive effect on both the UV-induced SF mobilization (Fig. 2A) and their displacement from irradiated sub-nuclear areas

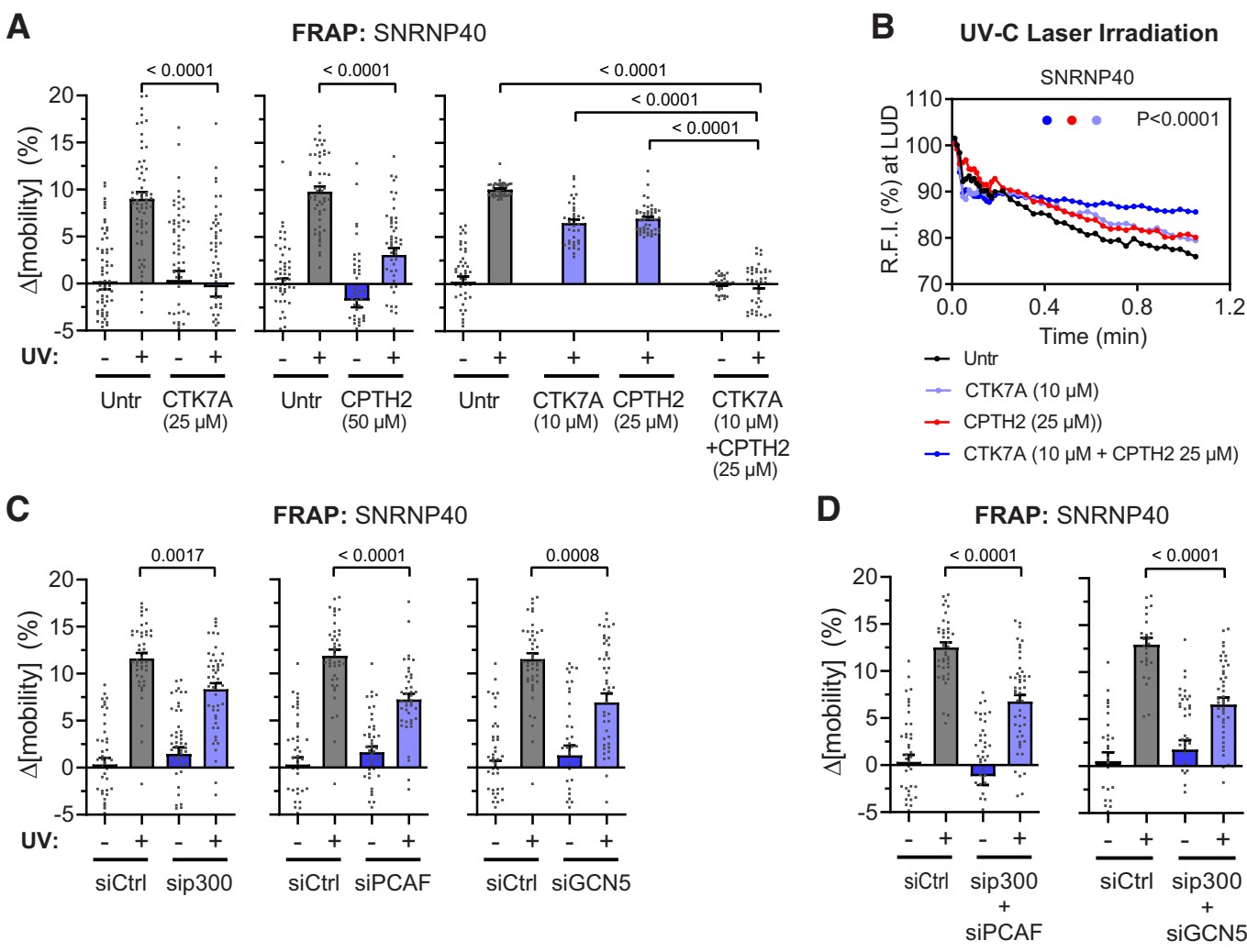

**Figure 2. HAT inhibition suppresses spliceosome displacement from damaged chromatin.**

(A, B) Pharmacological inhibition of histone acetyltransferases (HATs) attenuates the spliceosome response to DNA damage. SNRNP40, a marker of mature spliceosomes, was used to assess spliceosome mobility in quiescent HDFs treated for 2 h with the PCAF/p300 inhibitor Hydrazinocurcumin (CTK7A) or the GCN5 inhibitor CPTH2, alone or combined. (A) SNRNP40 mobility was measured by FRAP in non-irradiated and UV-irradiated (20 J/m², 30 min) quiescent HDFs. Plotted are changes in SNRNP40 mobility at 20 s post-bleaching normalized to untreated, non-irradiated controls. (B) Cells were pre-treated with suboptimal concentrations of CTK7A and CPTH2 prior to UV-C laser microbeam irradiation. Real-time displacement of SNRNP40-GFP from locally UV-induced DNA damage (LUD) regions was monitored by confocal microscopy. Plotted is fluorescence intensity at the irradiation site normalized to pre-irradiation levels. (C, D) Genetic inhibition of HATs suppresses UV-induced SNRNP40 mobilization. SNRNP40 mobility was analyzed by FRAP following siRNA-mediated knockdown of individual (C) or combined (D) HATs (PCAF, p300, GCN5) in quiescent HDFs. Graphs show SNRNP40 mobility at 20 s post-bleaching in UV-irradiated (20 J/m², 30 min) relative to control-transfected, non-irradiated cells. Data information: (A, C, D) Mean ± SEM of (left to right): (A) n = 67/68/66/64, 54/52/51/51, 47/51/36/46/33/41; (C) 44/44/51/47, 45/42/45/41, 47/46/38/49; and (D) 43/42/45/48, 28/26/40/45 cells; from three (A, C) or two (D) biological replicates. Brown–Forsythe and Welch's ANOVA with Games–Howell's multiple comparisons test. (B) Mean ± SEM of n = 14 (untreated), 27 (CTK7A), 30 (CPTH2), and 27 (CTK7A + CPTH2) cells from three biological replicates; repeated measures two-way ANOVA. Source numerical data and detailed statistical analysis are provided in the Source Data file. Source data are available online for this figure.

(Fig. 2B). Importantly, siRNA-mediated depletion of GCN5, PCAF or p300 by ~40–50% (Fig. EV2F,G) mimicked the effect of pharmacological inhibitors used at suboptimal concentrations (Fig. 2C,D).

**Transcription-dependent recruitment of p300 to UV-damaged chromatin**

To demonstrate a direct role of HATs in chromatin acetylation at the vicinity of TBLs, we examined the localization of endogenous

PCAF, p300, and GCN5 in response to DNA damage. We hypothesized that the responsible HATs acetylate chromatin in the vicinity of UV-induced photolesions and the subsequently formed R-loops, and that their localization depends on active transcription. Proximity ligation assays (PLA) in UV-irradiated quiescent HDFs revealed that PCAF, p300, and GCN5 localize within ~40 nm of UV-induced CPDs (Fig. 3A). Among the three, p300 showed the strongest PLA signal, detectable at doses as low as 5 J/m² in a dose-dependent manner (Fig. 3B). HAT recruitment to TBL-enriched subnuclear regions was independently confirmed by

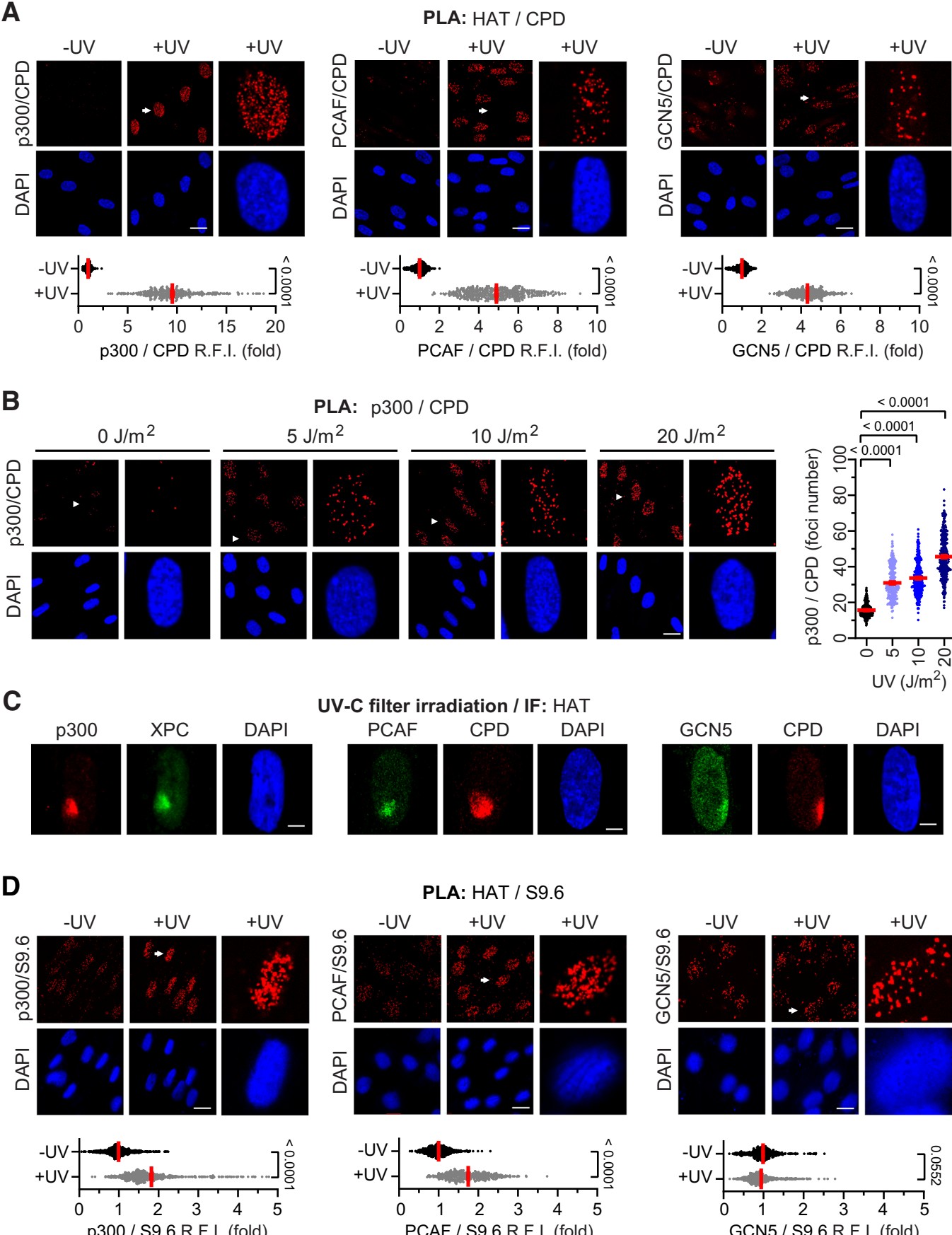

**Figure 3. HATs co-localize with UV-induced photolesions and R-loop-containing chromatin.**

(A, B) Representative Proximity Ligation Assay (PLA) images and quantifications showing HAT co-localization with UV-induced cyclobutane pyrimidine dimers (CPDs) in quiescent HDFs. (A) Plotted are PLA signal intensities between CPDs and p300, PCAF or GCN5 of UV-irradiated (30 J/m², 1 h) cells, relative to non-irradiated controls. (B) UV dose-dependent co-localization of p300 and CPDs. Plotted are PLA foci counts per nucleus across increasing UV doses. (A, B) Nuclei were stained with DAPI. Scale bars: 20 μm. Arrows indicate cells shown at higher magnification. (C) HAT recruitment to subnuclear UV-damaged regions detected by immunofluorescence in quiescent HDFs irradiated through isopore filters (80 J/m²). DNA damage sites were visualized by immunofluorescence detection of CPDs or the CPD-recognition protein XPC. Nuclei were stained with DAPI. Scale bars: 2 μm. (D) HAT co-localization with R-loop-containing chromatin. Representative PLA images and quantifications of PLA signals between HATs and RNA:DNA hybrids (detected by the S9.6 antibody) in quiescent HDFs. Plotted are PLA signal intensities in UV-irradiated (30 J/m², 1 h) versus non-irradiated controls. Data information: Mean ± SEM of (A) $n = 348/317$ (p300), 362/255 (PCAF), and 468/366 (GCN5); (B) 275/255/385/390; and (D) 634/618 (p300), 499/396 (PCAF), 405/568 (GCN5) cells. All data are from three biological replicates. Unpaired two-tailed $t$ test with Welch's correction (A, D), or Brown–Forsythe and Welch's ANOVA with Games–Howell's multiple comparisons test (B). Source numerical data, detailed statistical analysis and unprocessed images are provided in the Source Data file. Source data are available online for this figure.

immunofluorescence imaging using micropore-based UV irradiation (Fig. 3C). In non-irradiated cells, all three HATs were found in close proximity to R-loops, as detected by the S9.6 RNA:DNA hybrid-recognizing antibody (Boguslawski et al, 1986) (Fig. 3D). Following UV exposure, the PLA signal for p300 and PCAF increased, indicating enhanced association with DNA damage-induced R-loops (Fig. 4D). In contrast, GCN5's proximity to R-loops remained unchanged (Fig. 4D), suggesting a broader role for GCN5 in chromatin acetylation at R-loop-enriched regions across the genome, rather than a specific function at TBL-associated R-loops (Brand et al, 2001; Guo et al, 2011; Waters et al, 2015). This non-localized activity of GCN5 is further supported by the observation that, unlike p300 (and to a lesser extent PCAF), GCN5's co-localization with CPDs is not reduced by transcription inhibition (Fig. 4A,B).

Among the three HATs tested, p300 was the only one to exhibit strict dependence on active transcription for its recruitment to UV-damaged chromatin as shown by Proximity Ligation Assay (PLA) following transcriptional inhibition (Fig. 4A) and confirmed by immunoblotting of cross-linked chromatin (Fig. 4B). This contrasts with PCAF and GCN5, which localized to photolesions, and UV-irradiated chromatin regardless of transcriptional activity (Fig. 4A,B). Consistent with this, p300's proximity to R-loops was also transcription-dependent (Fig. 4C). These findings point to a distinct and specific role for p300 in the cellular response to transcription-blocking lesions (TBLs). We therefore propose that, in response to UV-irradiation, p300 is selectively recruited to lesion-stalled RNAPII, where it mediates localized chromatin acetylation and facilitates downstream signaling events, distinguishing it from more broadly acting chromatin modifiers such as PCAF and GCN5.

## R-loop-induced ATM activation depends on chromatin acetylation

Disruption of co-transcriptional splicing promotes R-loop formation between the protein-free unprocessed nascent RNA and the negatively supercoiled DNA upstream of RNAPII (Garcia-Muse and Aguilera, 2019; Li and Manley, 2005; Petermann et al, 2022; Salas-Armenteros et al, 2017; Stolz et al, 2019). Since HAT inhibition interferes with the UV-induced release of co-transcriptional spliceosomes, we hypothesized that it would also suppress the ensuing R-loop formation. To test this, we assayed the recruitment of GFP-tagged RNaseH1, an enzyme that degrades RNA in RNA:DNA hybrids (Wu et al, 2001; Tresini et al, 2015), to

microbeam irradiated sites. To avoid the risk of R-loop dissolution by active RNaseH1, a catalytically inactive but binding-competent mutant (D145N) of RNaseH1 (Wu et al, 2001) was used. The rapid accumulation of RNaseH1(D145N)-GFP at UV-irradiated sites was significantly reduced by HAT inhibition (Fig. 5A) indicating that, under these conditions, R-loop formation is suppressed. Furthermore, HAT inhibition abrogated the increase in UV-induced RNA:DNA hybrids, as detected by immunofluorescence using the S9.6 antibody (Fig. 5B). Notably, the S9.6 signal was transcription-dependent, confirming the antibody's specificity in detecting co-transcriptionally formed R-loops (Fig. 5B).

Given the critical role of co-transcriptional R-loop formation in TBL-induced ATM activation (Tresini et al, 2015) (Fig. EV3A–D), we argued that HAT inhibition would also impede this process. Indeed, both partial siRNA-mediated HAT depletion (Fig. EV3F,G) and pharmacological inhibition at concentrations that suppressed R-loop formation (Fig. 5A,B) abrogated UV-dependent ATM activation, as measured by immunofluorescence (Fig. 5C,D) and by immunoblotting (Fig. 5E) of auto-phosphorylated (active) ATM (Bakkenist and Kastan, 2003). Conversely, ATM was activated by pharmacologically induced chromatin acetylation (and ensuing spliceosome mobilization) by either the HDAC inhibitor TSA or the HMT inhibitors BIX01294 and GSK126, even in non-irradiated cells (Figs. 5F,G and EV3E). Thus, it appears that chromatin modifications that relax chromatin, whether triggered by TBLs or induced pharmacologically, are both necessary and sufficient to activate ATM.

Upon UV-irradiation, a subset of activated ATM molecules localizes proximal to CPDs (Fig. EV3F). Interestingly, we also observed a rapid accumulation to UVC-laser irradiated sites, of an inactive ATM truncation mutant (aa 2–1314) containing only its intact N-terminally located Spiral domain, involved in chromatin-association and substrate-recognition (Warren and Pavletich, 2022). HAT inhibition strongly suppressed this accumulation (Fig. EV3G), indicating that GFP-ATM(N-term) is only recruited to relaxed chromatin, presumably by TBL-associated R-loops formed at the irradiation sites.

## ATM-dependent phosphorylation of damaged chromatin

ATM, activated by DNA double-strand breaks (DSBs), catalyses the phosphorylation of the histone variant H2A.X (Burma et al, 2001), a critical chromatin modification involved in damage signalling and repair. We previously reported (Tresini et al, 2015), and also show here, that ATM activation by TBLs also leads to H2A.X

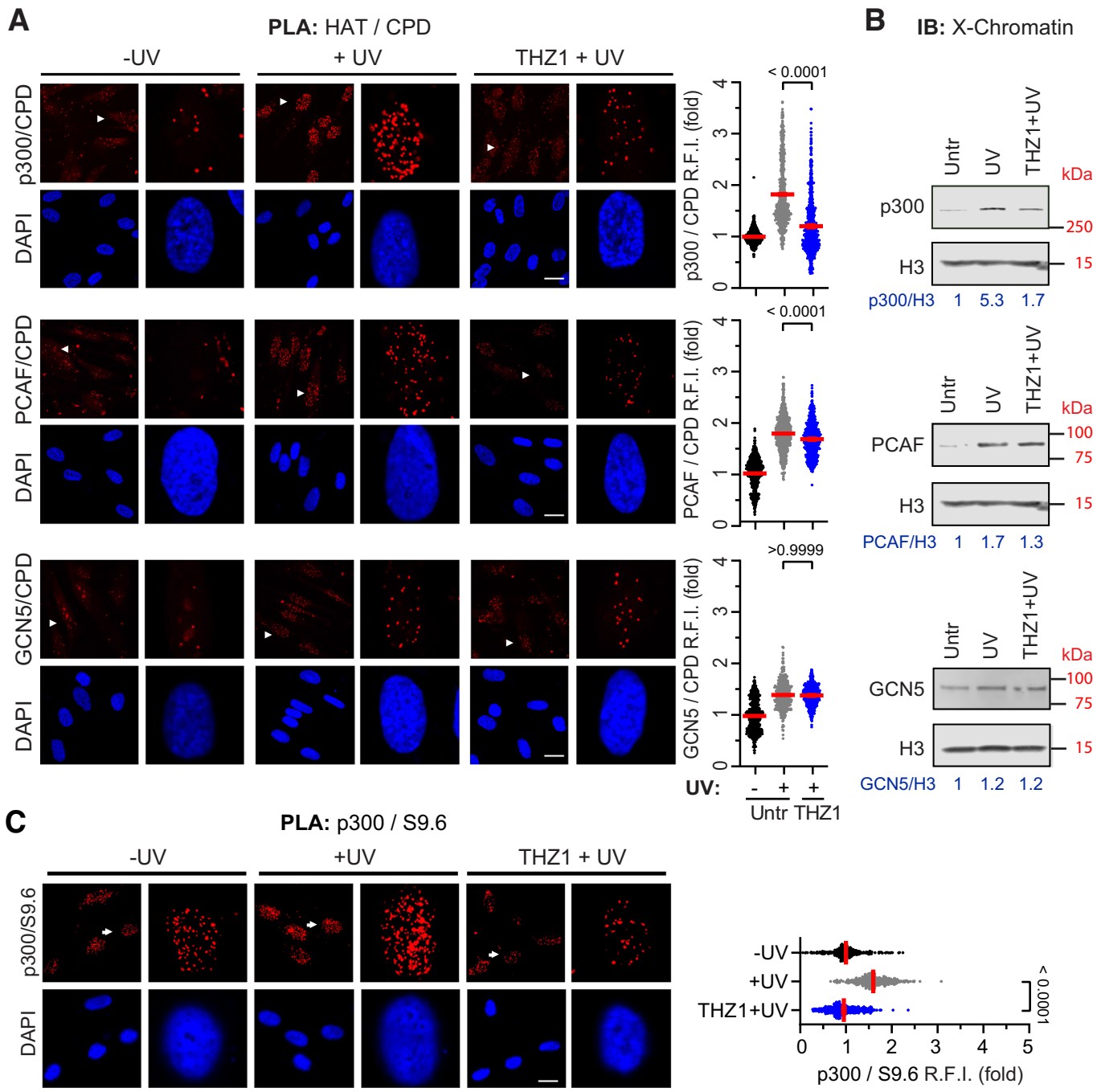

**Figure 4. UV-induced localization of p300 proximal to photolesions and R-loops is transcription dependent.**

(A) Representative PLA images and quantification showing co-localization of HATs (p300, PCAF, GCN5) with UV-induced CPDs in quiescent HDFs. Cells were pre-treated with transcription inhibitor THZ1 (1 μM, 3 h) as indicated prior to UV irradiation (20 J/m², 30 min). Nuclei were stained with DAPI. Arrows indicate cells shown at higher magnification. Scale bars: 20 μm. PLA signal intensities are plotted relative to untreated/non-irradiated controls. (B) Immunoblot analysis of cross-linked chromatin fractions showing UV-dependent HAT association in quiescent HDFs treated as in panel (A). Histone H3 serves as loading control. p300 and PCAF were detected on the same membrane and share the H3 control. (C) PLA detection of p300 co-localization with RNA:DNA hybrids (S9.6 antibody) in quiescent HDFs. Cells were untreated or treated with THZ1 (1 μM, 3 h) prior to UV-C irradiation. Nuclei were stained with DAPI. Arrows indicate cells shown at higher magnification. Scale bars: 20 μm. PLA signal intensities are shown relative to non-irradiated cells. Data information: Mean ± SEM of (left to right): (A) $n = 529/596/733$ (p300), 693/768/618 (PCAF), and 498/418/531 (GCN5) cells from four biological replicates; Brown–Forsythe and Welch's ANOVA with Games–Howell multiple comparisons test. (C) $n = 337/323/445$ cells from three biological replicates; one-way ANOVA with Bonferroni multiple comparisons test. Source numerical data, detailed statistical analysis and unprocessed images are provided in the Source Data file. Source data are available online for this figure.

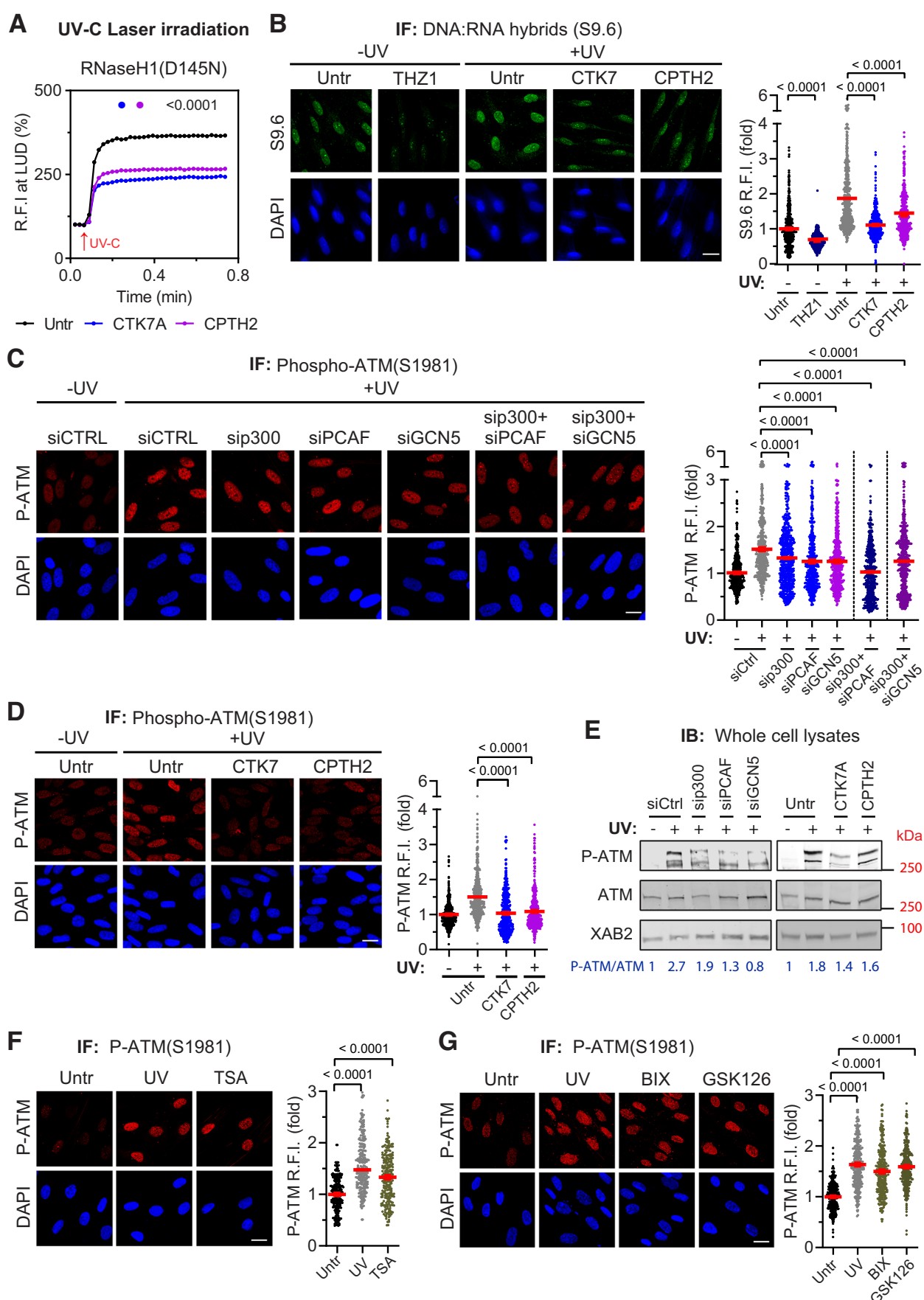

**Figure 5. Modulation of R-loop–dependent ATM activation by chromatin modifiers.**

(A) Pharmacological HAT inhibition suppresses recruitment of catalytically inactive GFP-tagged RNaseH1(D145N) to UV-C laser-irradiated chromatin. Quiescent HDFs were treated with the p300/PCAF inhibitor CTK7A (25 μM) or the GCN5 inhibitor CPTH2 (50 μM) for 2 h prior to UV-C laser microbeam irradiation. RNaseH1(D145N) accumulation kinetics at irradiated sites were monitored in real time by confocal microscopy. Plotted are fluorescence intensities at irradiated sites normalized to pre-irradiation levels. (B) Representative immunofluorescence (IF) images and quantification of RNA:DNA hybrid levels detected by S9.6 antibody in detergent-extracted nuclei of non-irradiated or UV-irradiated (40 J/m², 2 h) quiescent HDFs pre-treated with CTK7A, CPTH2, or the transcription inhibitor THZ1 (1 μM, 2 h). (C–E) HAT inhibition suppresses UV-induced ATM activation (40 J/m², 2 h) in quiescent HDFs. (C, D) Representative IF images and quantification of auto-phosphorylated (active) ATM. (E) Immunoblots of total and auto-phosphorylated ATM. XAB2 is shown as loading control. Prior to irradiation, cells were either (C, E) transfected with siRNAs targeting p300, PCAF, or GCN5 individually, or in combination, or (D, E) treated with HAT inhibitors for 2 h. (F, G) ATM activation in response to pharmacologically decompacted chromatin. Representative IF images and quantification of auto-phosphorylated ATM. Quiescent HDFs were UV-irradiated (20 J/m², 2 h) or treated with (F) the HDAC inhibitor Trichostatin A (1 μM, 2 h), or (G) histone methyltransferase (HMT) inhibitors BIX01294 (10 μM, 2 h) and GSK126 (10 μM, 2 h). Nuclei stained with DAPI. Scale bars: 20 μm. Plotted are normalized signal intensities relative to untreated or control-transfected, non-irradiated cells. Data information: (A) Mean ± SEM of $n = 36$ (untreated), 37 (CTK7A), and 37 (CPTH2-treated) cells from three biological replicates; repeated-measures two-way ANOVA. (B–D, F, G) Mean ± SEM of (left to right): (B) $n = 378/334/394/330/275$; (C) 506/504/522/484/509/524/448; (D) 575/496/489/414; (F) 468/450/415; and (G) 290/334/283/314; all from three biological replicates. Brown–Forsythe and Welch's ANOVA with Games–Howell's multiple comparisons test. Source numerical data, detailed statistical analysis, and unprocessed images are provided in the Source Data file. Source data are available online for this figure.

phosphorylation (Fig. EV3H,I,J). However, unlike in response to DSBs, phosphorylated H2A.X does not accumulate in nuclear foci but instead exhibits a pan-nuclear distribution (Fig. EV3H,I), confirming that this is a DSB-independent event (Hanasoge and Ljungman, 2007; Marti et al, 2006; Tresini et al, 2015). We expand on this observation by demonstrating that, under DNA damage-induced transcription stress, ATM-regulated chromatin phosphorylation also includes the Ser10 phosphorylation of histone H3 (H3S10ph) (Figs. 6A,B and EV5A). In initial screens of UV-induced histone PTMs in non-replicating HDFs, we observed a clear increase in H3S10ph (Fig. EV1A). In agreement with prior reports describing H3S10ph enrichment at sites of aberrant R-loops (Castellano-Pozo et al, 2013; Garcia-Pichardo et al, 2017), we found elevated levels of this chromatin mark proximal (within 40 nm) to UV-induced CPDs (Fig. EV4A) and RNA:DNA hybrids (Fig. EV4B). Notably, the UV-induced H3S10 phosphorylation also depends on active transcription (Figs. 6C,D and EV4C) and R-loop levels (Fig. 6E,F) indicating that this modification is stimulated by TBL-induced R-loops. Transcription inhibition or overexpression of wild-type RNaseH1—but not a catalytically dead mutant (D145N)—suppressed UV-induced H3S10ph (Fig. 6F), while RNaseH1 knock-down enhanced it (Fig. 6E) indicating that TBL-induced R-loops trigger this modification. On the premise that H3S10ph deposition "marks" chromatin regions containing aberrant R-loops (Garcia-Pichardo et al, 2017), we asked whether it mechanistically contributes to ATM activation. To test this we used inhibitors targeting kinases previously implicated in stimulus dependent H3S10 phosphorylation (Komar and Juszczynski, 2020; Zhong et al, 2001). These experiments identified Mitogen- and Stress-activated Protein Kinase 1 (MSK1) and its upstream activator, p38 Mitogen-Activated Protein Kinase (p38 MAPK), as the main regulators of H3S10ph following UV irradiation (Figs. EV4D,E and EV5A). Pharmacological inhibition of either MSK1 or p38 MAPK abolished UV-induced H3S10ph but not of activated ATM (Figs. EV4E and EV5A) suggesting that ATM activation does not rely on R-loop demarcation by H3S10ph. Surprisingly, inhibition or depletion of ATM supressed H3S10ph (Figs. 6A,B and EV5A) consistent with an important role of ATM. DNA:RNA immunoprecipitations (DRIP) using the S9.6 antibody on native chromatin from UV irradiated quiescent HDFs, confirmed the presence of H3S10ph at R-loop-containing regions and its attenuation upon ATM inhibition (Fig. 6G).

This influence of ATM on H3S10 phosphorylation indicates either a synergism of the p38 MAPK/MSK1 and ATM pathways, or alternatively,

that ATM functions mechanistically upstream of MSK1. ATM has previously been linked to the activation of TAK1 (Transforming growth factor β-activated kinase 1) and TAOK (Thousand and one amino acid kinases), two MAP3Ks involved in MAP2K-catalyzed p38 MAPK phosphorylation during cellular stress (Canovas and Nebreda, 2021; Raman et al, 2007; Yang et al, 2011) (Fig. EV4D). ATM silencing or pharmacological inhibition prior to UV-irradiation partly suppressed both p38 and MSK1 phosphorylation (Figs. 7A,B and EV5B,C) which could account for the reduced H3S10ph levels we observed (Figs. 6A and EV5A,C). Importantly, ATM inhibition also reduced the dual phosphorylation of p38's activation-loop Thr–Gly–Tyr motif (Thr180/Tyr182 of the p38α isoform) (Figs. 5A and EV5B,C). Since this dual phosphorylation event —required for full p38 activation— is catalyzed by dual-specificity Ser/Thr-Tyr kinases, and not by the Ser/Thr-specific kinase ATM, these findings strongly argue against a direct action of ATM on p38 itself. Instead, they suggest that ATM acts upstream, most likely through phosphorylation of p38-activating MAP3Ks. TAOK activity has been shown to depend on direct ATM phosphorylation and to be critical for p38 activation in response to UV and ionizing radiation (Raman et al, 2007). In that study, ATM target sites were mapped to specific residues on TAOK1 and TAOK3, providing strong support for the mechanistic significance of this phosphorylation event in p38 activation.

The H3S10ph mark has been implicated in diverse cellular processes, including the coordination of stress responses and transcription reprogramming (Komar and Juszczynski, 2020; Sawicka and Seiser, 2012). Meta-analysis of our RNA-Seq data of ATM-dependent, UVC-upregulated transcripts (Data ref: Tresini et al, 2015), revealed that several H3S10ph-regulated genes encoding stress-response transcription factors (e.g. c-myc, FOSL1, c-JUN, JUNB, EGR1) (Komar and Juszczynski, 2020), also require ATM-activity for proper induction. Hence, in line with the proposed functions of H3S10ph in "marking" aberrant R-loop-containing chromatin and, independently, in coordinating stress responses, ATM likely contributes to both processes by indirectly influencing H3S10ph levels. Interestingly, pharmacological suppression of p38 MAPK, MSK1, or ATM reduced UV-viability in non-replicating HDFs (Fig. 7C). While these signalling kinases have diverse downstream effects, the established role of H3S10ph in regulating chromatin dynamics and stress-induced transcriptional responses (Komar and Juszczynski, 2020) suggests that the reduced viability upon their inhibition may be partly due to H3S10ph suppression.

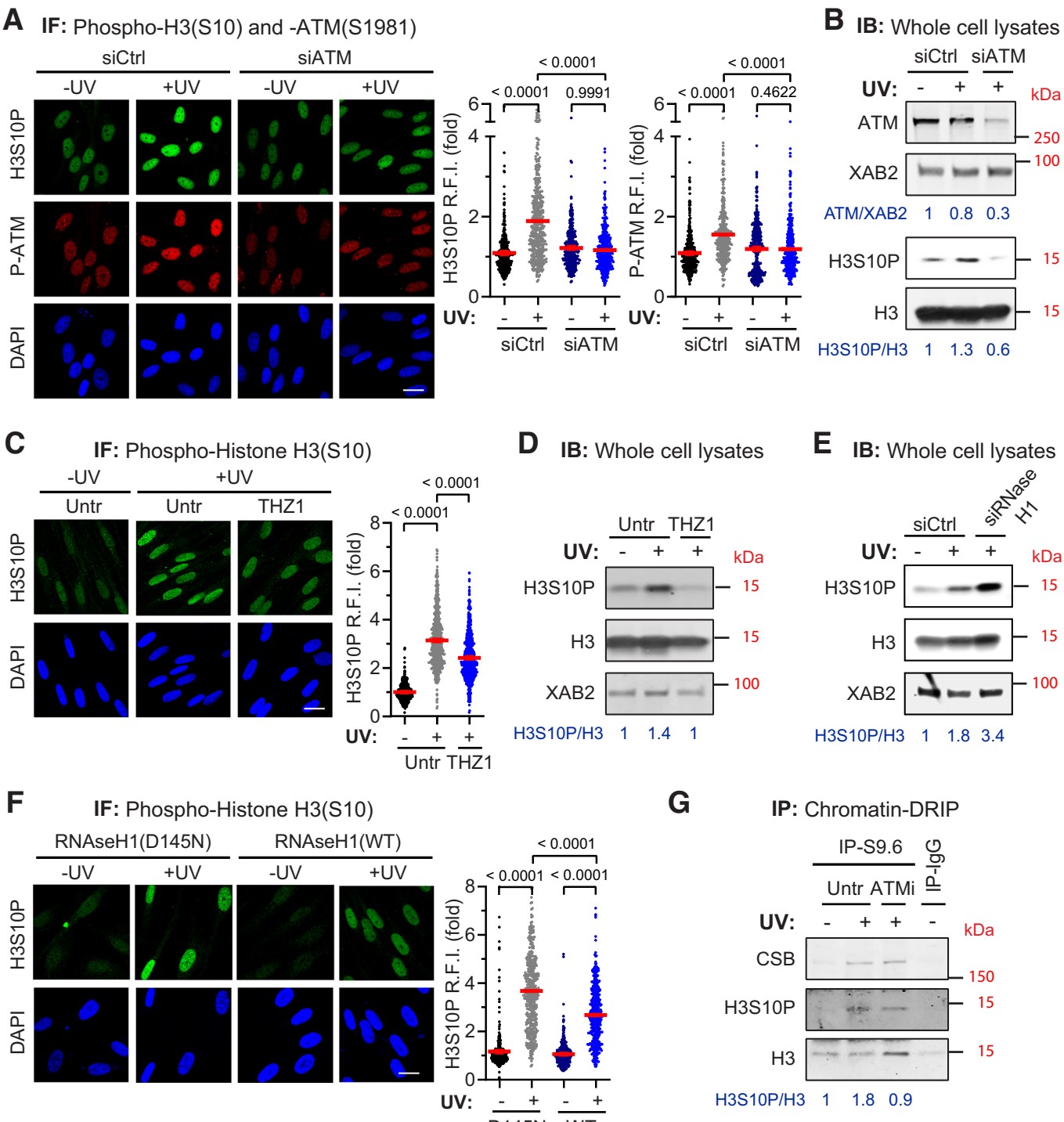

## Discussion

Stalling of transcription elongation complexes at DNA helix-distorting lesions initiates a series of events suppressing transcription both locally, at the site of stalled RNAPII, but also genome-wide (Lans et al, 2019; Steurer et al, 2022). Simultaneously, DDR mechanisms, partly coordinated by ATM, reprogram transcription (Tresini et al, 2015) to assist in DNA damage recovery, or trigger

senescence and apoptosis when damage levels disrupt homeostasis (Lans et al, 2019).

Here we show that in response to TBLs, the pathway leading to ATM activation initiates by transient chromatin acetylation occurring at the vicinity of the lesion. Under unperturbed conditions, HATs regulate chromatin acetylation at promoters/enhancers and track with transcription elongation complexes to relax chromatin ahead of transcribing RNAPII (Talbert and Henikoff, 2021). We propose that,

**Figure 6.   R-loop and ATM-dependent regulation of histone H3(S10) phosphorylation.**

(A, B) Reduced UV-induced histone H3 serine 10 phosphorylation [H3(S10)P] upon ATM depletion. Quiescent HDFs transfected with control or ATM-targeting siRNAs were mock-treated or UV-irradiated (40 J/m$^2$, 2 h). (A) Representative immunofluorescence images and quantification of phosphorylated H3(S10) and auto-phosphorylated ATM. (B) Immunoblots showing ATM depletion efficiency and H3(S10)P levels. Histone H3 and XAB2 serve as loading controls. (C–F) Optimal UV-induced H3(S10) phosphorylation depends on co-transcriptional R-loop formation. (C, D) Quiescent HDFs treated with the transcription inhibitor THZ1 (1 μM, 4 h) prior to UV irradiation (40 J/m$^2$, 2 h) show suppressed H3(S10)P. (C) Representative IF images and quantification of H3(S10)P signal intensities. (D) Immunoblots for H3(S10)P and loading controls Histone H3 and XAB2. (E) Immunoblots showing increased H3(S10)P upon RNaseH1 depletion in UV-irradiated (40 J/m$^2$, 2 h) quiescent HDFs. (F) Representative IF images and quantification of H3(S10)P signal intensities in unperturbed and UV-irradiated HDFs expressing mCherry-tagged wild-type RNaseH1 or catalytically inactive D145N mutant. Nuclei stained with DAPI. Scale bars: 20 μm. (G) Histone H3(S10) phosphorylation in proximity to UV-induced R-loops is partly suppressed by ATM inhibition. DNA:RNA immunoprecipitation (DRIP) was performed using the S9.6 antibody on chromatin fractions from unperturbed or UV-irradiated (40 J/m$^2$, 2 h) HDFs treated with or without the ATM inhibitor KU55933 (10 μM, 2 h). Immunoblots show total and phosphorylated H3(S10). The transcription-coupled nucleotide excision repair (TC-NER) protein CSB is shown as a marker of R-loop proximity to transcription-blocking lesions (TBLs). Data information: Mean ± SEM of (left to right): (A) $n = 463/411/361/400$ for both H3S10P and P-ATM; (C) $n = 696/698/646$; and (F) $n = 444/423/463/469$; all from three biological replicate experiments. Brown–Forsythe and Welch's ANOVA with Games–Howell's multiple comparisons test. Source numerical data, detailed statistical analysis, and unprocessed images are provided in the Source Data file. Source data are available online for this figure.

upon RNAPII stalling at TBLs, p300 and to a lesser extent PCAF HATs, are redirected to cooperatively acetylate and relax chromatin upstream of the stalled polymerase (Fig. 7D). The altered chromatin architecture stimulates spliceosome-release from lesion-stalled elongation complexes, presumably to enable RNAPII displacement from the lesion allowing subsequent repair by TC-NER (Mullenders, 2015). However, chromatin relaxation and spliceosome eviction also facilitate formation of R-loops (Tresini et al, 2015) between unspliced, snRNP-free nascent RNA and the unpaired DNA strand in the relaxed, "hyper-accessible" chromatin (Salas-Armenteros et al, 2017; Sanz et al, 2016), ultimately leading to ATM activation. Notably, forced chromatin acetylation (Bakkenist and Kastan, 2003) (Figs. 5G and EV3E), disruption of co-transcriptional RNA splicing (Tresini et al, 2015) or increased R-loop levels (Tresini et al, 2015), can all activate ATM even in absence of DNA damage. These findings are in line with a broad role of ATM in orchestrating the cellular response to various forms of transcription and/or chromatin stress (Bakkenist and Kastan, 2003, 2015; Shiloh, 2014; Tresini et al, 2015).

Upon its activation by TBLs, ATM orchestrates the cellular response through diverse mechanisms, ranging from spliceosome remodelling (Tresini et al, 2015) to the activation of key regulatory proteins, such as transcription factors (e.g. p53), and signalling kinases (e.g. Chk2) (Lee and Paull, 2021; Shiloh and Ziv, 2013). ATM also influences chromatin conformation via several mechanisms, two of which are explored in this manuscript: (i) direct phosphorylation of the histone variant H2A.X (Tresini et al, 2015) and (ii) indirect activation of the p38 MAPK/MSK1 pathway, which catalyses histone H3S10 phosphorylation. Although neither modification directly drives chromatin relaxation, both have been shown to contribute to chromatin decompaction indirectly (Georgoulis et al, 2017), (Komar and Juszczynski, 2020), potentially contributing to ATM's regulation of spliceosome dynamics.

Phosphorylation of histone H3S10 underpins two structurally opposed processes: chromatin compaction leading to transcriptional repression in mitosis (Komar and Juszczynski, 2020; Sawicka and Seiser, 2012) and in R-loop-containing regions (Castellano-Pozo et al, 2013; Garcia-Pichardo et al, 2017; Skourti-Stathaki et al, 2014), and chromatin relaxation, promoting expression of stress-response transcription factors (Komar and Juszczynski, 2020; Perez-Cadahia et al, 2009; Sawicka and Seiser, 2012). These apparentlly opposing effects on gene expression are likely influenced by the genomic location of P deposition and the context-dependent deposition of additional histone marks in its

vicinity (Komar and Juszczynski, 2020; Sawicka and Seiser, 2012). It is thus feasible that ATM-controlled phosphorylation of histone H3S10 may also exert opposing effects at different genomic locations. In the gene body, it may promote compaction to insulate R-loop containing chromatin from upstream elongating transcription complexes. Conversely, in promoters of stress-response genes it may establish a permissive chromatin environment for gene expression.

The physiological significance of the chromatin-R-loop-ATM axis is also supported by the phenotype of a murine model with ATM-deficiency mimicking the severe human neurological disorder Ataxia Telangiectasia (AT). Animals with combined *ATM* and Aprataxin (*Aptx*) deficiencies, which cause high basal levels of DNA damage and genome instability recapitulate the human AT syndrome and develop neurodegeneration and debilitating ataxia. At the molecular level $Atm^{R35X/R35X}$; $Aptx^{-/-}$ mice manifest aberrant messenger RNA splicing and R-loop formation in the cerebellum (Kwak et al, 2021). Notably, the observed changes are specific to Purkinje cells and associate with unique (open) chromatin conformation at the affected gene loci.

Interestingly, aberrant TBL-repair, spliceosome regulation, R-loop accumulation and ATM deficiency have all been linked to neurodegenerative conditions with overlapping but tissue/neuronal cell type-specific pathologies, ranging from inherited genome instability disorders to aging associated neuropathology (Barzilai et al, 2017; Lans et al, 2019; Lee and Paull, 2021; Mitiagin and Barzilai, 2023; Richard and Manley, 2017; Shiloh, 2020). The reciprocal influence between ATM and chromatin dynamics, which vary among different neuronal cell types (Lee et al, 2018), may offer a plausible explanation for how disruptions in this pathway could impact neuropathology in conditions with both common and distinct clinical manifestations.

# Methods

**Reagents and tools table**

| Reagent/resource | Reference or source | Identifier or catalog number |
|---|---|---|
| **Experimental models** | | |
| VH10 (*H. sapiens*) | PMID: 6096694 | RRID:CVCL_RW72 |
| C5Ro (*H. sapiens*) | PMID: 3803387 | RRID:CVCL_ZP35 |
| RPE1 (*H. sapiens*) | ATCC (CRL-4000) | RRID:CVCL_4388 |

| Reagent/resource | Reference or source | Identifier or catalog number |
|---|---|---|
| C5RoT-SNRNP40-GFP | Tresini et al, 2015 | N/A |
| C5RoT-SF3a1-GFP | Tresini et al, 2015 | N/A |
| C5RoT-U1A-GFP | Tresini et al, 2015 | N/A |
| C5RoT-PRP8-GFP | Tresini et al, 2015 | N/A |
| C5RoT PRPF3-GFP | Tresini et al, 2015 | N/A |
| VH10-GFP-RNaseH1(D145N) | Tresini et al, 2015 | N/A |
| VH10-mCherry-RNaseH1 (dox) | This study | N/A |
| VH10- mCherry RNaseH1(D145N) (dox) | This study | N/A |
| VH10-GFP-ATM(2-1314 aa) | This study | N/A |
| *Gryphon A* packaging cell line (*H. sapiens*) | Allele Biotechnology | ABP-RVC-10002C |
| HEK-293 (*H. sapiens*) | ATCC | CRL-1573 |
| S9.6 HB-8730™ (*M. musculus*) | ATCC | HB-8730 |
| RPE1-CSB-mScarlet-I knock-in | van Sluis M et al, 2024 | N/A |
| **Recombinant DNA** | | |
| pcDNA3.1(+)Flag-His-ATM wt | Addgene | RRID:Addgene_31985 |
| RNase H1 (RNASEH1) (NM_002936) Human Untagged Clone | Origene | SC319446 |
| pLHCX-SNRNP40-GFP | Tresini et al, 2015 | N/A |
| pLHCX-SF3a1-GFP | Tresini et al, 2015 | N/A |
| pLHCX- U1A-GFP | Tresini et al, 2015 | N/A |
| pLHCX- PRP8-GFP | Tresini et al, 2015 | N/A |
| pLHCX- PRPF3-GFP | Tresini et al, 2015 | N/A |
| pLHCX-RNaseH1(D145N)-GFP | Tresini et al, 2015 | N/A |
| pLenti(Dox)-mCherry-RNAseH1 | This study | N/A |
| pLenti(Dox)-mCherry-RNAseH1(D145N) | This study | N/A |
| pLenti(Dox)-GFP-ATM(2-1314 aa) | This study | N/A |
| **Antibodies** | | |
| Mouse anti-phospho-ATM(Ser1981) mAb clone 10H11.E12 | Merck | Cat# 05-740 |
| Rabbit anti-ATM | Abcam | Cat# ab199726 |
| Mouse anti-CPD mAb clone TDM-2 | COSMO BIO USA | Cat# CAC-NM-DND-001 |
| Rabbit anti-phospho-Histone H3 (Ser10) | Millipore | Cat# 06-570 |
| Mouse anti-Histone H3 | Cell Signaling Technology | Cat# 1B1B2 |
| Mouse anti-a-Tubulin mAb clone B-5-1-2 | Sigma-Aldrich | Cat# T5168 |
| Rabbit anti-KAT2B/PCAF | Abcam | Cat# ab12188 |
| Rabbit anti-KAT3B/p300 | Abcam | Cat# ab10485 |
| Mouse anti-p300 mAb clone F-4 | Santa Cruz Biotechnology | Cat# sc-48343 |
| Mouse anti-GCN5 mAb clone A-11 | Santa Cruz Biotechnology | Cat# sc-365321 |
| Mouse anti-GFP clones 7.1 and 13.1 | Roche | Cat# 11814460001 |
| Mouse anti-XAB2/HCN5 mAb clone S9 | Santa Cruz Biotechnology | Cat# sc-271037 |
| Rabbit anti-phospho-MSK1 | Cell Signaling Technology | Cat# 9595 |
| Rabbit anti-Phospho-H2A.X (S139) mAb | Abcam | Cat# ab22551 |
| Mouse anti-phosphorH2A.X(S139) mAb clone JBW301 | Millipore /MERCK | Cat# 05-636 |
| Rabbit anti-H2A.X | Cell Signaling Technology | Cat# 2595 |
| Rabbit anti-Histone H3(acetyl K9, + K14, + K18, + K23, + K27) | Abcam | Cat# ab47915 |
| Mouse anti-H3K9Me2 mAb | Abcam | Cat# ab1220 |
| Rabbit anti-H3K27Me3 | Abcam | Cat# ab272165 |
| Mouse anti-phosphpo-p38 (Thr180/Tyr182) mAb clone 28B10 | Cell Signaling Technology | Cat# 9216 |
| Rabbit anti-p38 | Cell Signaling Technology | Cat# 9212 |
| Rabbit anti-XPC | Bethyl Labs | Cat# A301-122A |

| Reagent/resource | Reference or source | Identifier or catalog number |
|---|---|---|
| Rabbit anti-CSB/ERCC6 | Antibodies on line | Cat# ABIN2855858 |
| Rat anti-RNA Pol II Ser2-P mAb clone 3E10 | ChromoTek | Cat# 3E10 |
| Goat anti-rabbit Alexa Fluor 488 | Invitrogen | Cat# A11034 |
| Goat anti-mouse Alexa Fluor 488 | Invitrogen | Cat# A11001 |
| Goat anti-rabbit Alexa Fluor 594 | Invitrogen | Cat# A21207 |
| Goat anti-mouse Alexa Fluor 594 | Invitrogen | Cat# A11032 |
| Goat anti-rat CF™ IRDye 680 | Sigma-Aldrich | Cat# sab4600480 |
| Goat anti-rabbit CF™ IRDye 770 | Sigma-Aldrich | Cat# sab4600215 |
| Goat anti-mouse CF™ IRDye 680 m | Sigma-Aldrich | Cat# sab4600199 |
| Goat anti-rabbit CF™ IRDye 680 | Sigma-Aldrich | Cat# sab4600200 |
| Goat anti-mouse CF™ IRDye 770 | Sigma-Aldrich | Cat# sab4600214 |
| Mouse S9.6 mAB clone HB-8730 | Purified in this study from ATCC HB-8730 hybridoma cell line | N/A |
| **Oligonucleotides and other sequence-based reagents** | | |
| siGENOME Non-Targeting siRNA #2 | Horizon Discovery/ Dharmacon™ | Cat# D-001210-02-20 |
| ON-TARGET plus Human CREBBP (1387) siRNA SMARTpool | Horizon Discovery/ Dharmacon™ | Cat# L-003477-00-0005 |
| ON-TARGET plus Human EP300 (2033) siRNA SMARTpool | Horizon Discovery/ Dharmacon™ | Cat# L-003486-00-0010 |
| ON-TARGETplus Human RNASEH1 (246243) siRNA - SMARTpool | Horizon Discovery/ Dharmacon™ | Cat# L-012595-01-0010 |
| ON-TARGETplus Human KAT2A (2648) siRNA - SMARTpool | Horizon Discovery/ Dharmacon™ | Cat# L-009722-02-0010 |
| siGCN5 (siKAT2A) (CCAUUUGAGAAACCUAAUAdTdT) | Horizon Discovery/ Dharmacon™ | N/A |
| ON-TARGETplus Human KAT2B (8850) siRNA - SMARTpool | Horizon Discovery/ Dharmacon™ | Cat# L-005055-00-0010 |
| siPCAF (siKAT2B) (GGUGGUAUCUGUUUCCGUAdTdT) | Horizon Discovery/ Dharmacon™ | N/A |
| siATM (GCCUCCAGGCAGAAAAAGAdTdT) | Horizon Discovery/ Dharmacon™ | N/A |
| **Chemicals, enzymes and other reagents** | | |
| Cordycepin | Sigma-Aldrich | Cat# C3394 |
| Flavopiridol hydrochloride hydrate | Sigma-Aldrich | Cat# F3055 |
| THZ1-2HCL | Selleckchecm | Cat# S5749 |
| Trichostatin A | Sigma-Aldrich | Cat# T8552 |
| Anacardic Acid | Millipore / MERCK | Cat# 172050 |
| Histone Acetyl Transferase Inhibitor VII, CTK7A | Calbiochem | Cat# 382115 |
| Cyclopentylidene-[4-(4-chlorophenyl)thiazol-2-yl)hydrazine (CPTH2) | Sigma-Aldrich | Cat# c9873 |
| SB203580 | Cell Signaling Technology | Cat# 5633 |
| SB747651A dihydrochloride | Tocris Bioscience | Cat# 4630 |
| BIX01294 Trihydrochloride hydrate | Sigma-Aldrich | Cat# B9311 |
| GSK126 | Selleckchem | Cat# S7061 |
| KU-55933 | Selleckchem | Cat# S1092 |
| Pladienolide B | Santa Cruz Biotechnology | Cat# Sc-391691 |
| Doxycycline | Sigma-Aldrich | Cat# D9891 |
| Benzonase® Nuclease | Millipore/MERCK | Cat# E1014 |
| Nuclease micrococcal (MNase) | Sigma-Aldrich | Cat# N5386 |
| Proteinase K | Roche/MERCK | Cat# RPROTKSOL-RO |
| Lipofectamine™ RNAiMAX Transfection Reagent | Invitrogen/Thermo-Fischer Scientific | Cat# 13778075 |
| FuGENE® 6 Transfection Reagent | Promega | Cat# E2691 |
| 5-Ethynyl-uridine (5-EU) | Axxora | Cat# JBS-CLK-N002 |
| Ascorbic Acid | Sigma-Aldrich | Cat# 09198 |
| Atto 594 Azide | Atto Tec | Cat# AD594-105 |
| CuSO4 * 5 H2O | Sigma-Aldrich | Cat# A0278 |

| Reagent/resource | Reference or source | Identifier or catalog number |
|---|---|---|
| EDTA-free Protease Inhibitor Cocktail | Roche | Cat# 11836170001 |
| Phosphatase Inhibitor Cocktail 3 | Sigma-Aldrich | Cat.# P0044 |
| Phosphatase Inhibitor Cocktail 2 | Sigma-Aldrich | Cat# P5726 |
| Bovine Serum Albumin (BSA) | Sigma-Aldrich | Cat# A3294 |
| Formaldehyde solution | Sigma-Aldrich | Cat# 47608 |
| Methanol | Honeywell | 32213 |
| Triton™ X-100 | Sigma-Aldrich | Cat# T8787 |
| Trizma® base | Sigma-Aldrich | Cat# T6066 |
| TWEEN® 20 | Sigma-Aldrich | Cat# P1379 |
| Nonidet P 40 Substitute (NP40) | Fluka | Cat# 74385 |
| 2 × Laemmli sample buffer | Sigma-Aldrich | Cat# S3401 |
| Dulbecco's modified Eagle's medium (DMEM) | Gibco™ | Cat# 11965084 |
| Ham's F-10 Nutrient Mix | Gibco™ | Cat# 11550043 |
| PFHM-II Protein-Free Hybridoma Medium | Gibco™ | Cat# 12040-077 |
| Fetal Bovine Serum | Capricorn | Cat# FBS-12A |
| Penicillin/streptomycin (PS) | Sigma-Aldrich | Cat# P0781 |
| Opti-MEM™ I Medium | Gibco™ | Cat# 31985047 |
| Dulbecco's Phosphate Buffered Saline (PBS) | Sigma-Aldrich | Cat# D8537 |
| Saline-Sodium Citrate (SSC) buffer (20X) | Thermo-Fischer Scientific | Cat# J60839.K3 |
| **Software** | | |
| ZEISS ZEN 2012 SP5 (version 14.06.201) | Carl Zeiss | N/A |
| LAS AF (version 2.7.4.10100) | Leica Microsystems | N/A |
| LAS X (version 3.5.6.21594) | Leica Microsystems | N/A |
| Fiji ImageJ (version 1.52p) | https://imagej.net/ij/index.html | N/A |
| ImageQuant™ TL v8.0.0. | Cytiva | N/A |
| Image Studio Lite (version 5.2.5) | LI-COR Biosciences | N/A |
| Prism GraphPad (version 9.4.0) | GraphPad Software Inc. https://www.graphpad.com/ | N/A |
| **Other** | | |
| TUV lamp (UV-C) | Phillips | N/A |
| Isopore membrane filter 8 µm pores | Millipore | Cat# TETP04700 |
| Isopore membrane filter 5 µm pores | Millipore | CAT#TMTP04700 |
| Bioruptor Sonicator | Diagenode | N/A |
| 4–15% Mini-PROTEAN® TGX™ Precast Protein Gels, 15-well | BioRad | Cat# 4561086 |
| 4–15% Mini-PROTEAN® TGX™ Precast Protein Gels, 12-well | BioRad | Cat# 4561085 |
| HiTrap™ MabSelect SuRe™ column | GE Healthcare | Cat# 29-0491-04 |
| Histone Extraction Kit | Abcam | Cat# ab113476 |
| Histone H3 Modification Multiplex Assay Kit | Abcam | Cat# ab185910 |
| Histone H4 Modification Multiplex Assay Kit | Abcam | Cat# ab185914 |
| Pierce™ BCA Protein Assay Kit | Thermo-Fischer Scientific | Cat# 23227 |
| QIAquick PCR Purification Kit | Qiagen | Cat# 28104 |
| AlamarBlue™ Cell Viability Reagent | Invitrogen/Thermo-Fischer Scientific | Cat# A50100 |
| ChromoTek RFP-Trap® Agarose | Proteintech | Cat# rta |
| ChromoTek Agarose binding control beads | Proteintech | Cat# bab-20 |
| Duolink®proximity ligation assay(PLA®) Kit | Sigma-Aldrich/MERCK | Cat# DUO92101 |
| Duolink® In Situ Mounting Medium with DAPI | Sigma-Aldrich/MERCK | Cat# DUO82040 |
| 4′,6-diamidino-2-phenylindole (DAPI) | Sigma-Aldrich | Cat# D9542 |

| Reagent/resource | Reference or source | Identifier or catalog number |
|---|---|---|
| Aqua-Poly/Mount | Polysciences, Inc. | Cat# 18606-20 |
| Amersham™ Protran® Western blotting membranes, nitrocellulose | Amersham/Merc | Cat# GE10600002 |
| Immobilon® -P PVDF Membrane | Millipore/Merck | Cat# IPVH00010 |
| ÄKTA start | CYTIVA (https://www.cytivalifesciences.com/en/us) | Cat# 29022094 |
| Promega Glomax® Multimode reader | Promega | N/A |
| LI-COR Odyssey CLx Imaging System | Li-COR | N/A |
| Leica TCS SP5 AOBS laser scanning confocal microscope | Leica Microsystems | N/A |
| Zeiss LSM700 laser-scanning confocal microscope | Zeiss | N/A |

## Methods and protocols

### Cell lines, cell culture conditions and chemical treatments

Parental cell lines used in this study were: TERT immortalized VH10 human foreskin fibroblasts and C5Ro adult female donor-derived human dermal fibroblasts. Cell lines generated to ectopically express fusion proteins were as follows: C5RoT stably expressing SNRNP40-GFP, SF3a1-GFP, U1A-GFP or PRPF3-GFP; VH10T expressing doxycycline (Dox)-inducible wild-type (WT) mCherry-RNaseH1 or the catalytically inactive mutant mCherry-RNaseH1(D145N), or stably expressing the RNaseH1 catalytically inactive mutant RNaseH1(D145N)-GFP; and VH10T stably expressing the GFP-tagged N-terminal (2–1314 aa) of ATM. For retrovirus transduction the amphotropic retroviral packaging cell line Gryphon A (genetically modified HEK 293 embryonic kidney cells) was purchased from Allele Biotechnology and for S9.6 antibody purification the HB-8730 mouse hybridoma cell line was purchased from ATCC. For expression of GFP-and mCherry fusion proteins cDNAs were subcloned and stably expressing cell lines were generated as previously described (Tresini et al, 2015). In all subcloned RNaseH1 cDNAs the mitochondrial localization signal (amino acids 1–28) was omitted. Female retinal pigmented epithelial RPE-1 CSB–mScarlet-I Knock-in were a kind gift of Drs M. van Sluis and J. Martijn (van Sluis M et al, 2024). Cell lines were not authenticated. All cell lines were routinely tested for mycoplasma and were negative. Cells were cultured at 37 °C in 5% $CO_2$ in a humidified incubator and maintained in Ham's F10 (C5Ro, VH10) or DMEM growth medium, supplemented with 10% fetal bovine serum (FBS) and 1% penicillin–streptomycin. Cells were synchronized in quiescence (G0) by contact inhibition and serum deprivation for 48–72 h in serum-free growth medium. Chemicals used for treatments, listed in the Reagents and Tools table, were added directly in the serum deprivation medium. Drug concentrations and treatment times are indicated in the figure legends. Doxycycline was used for inducible cDNA expression.

### Antibodies

Commercially available antibodies used for Immunofluorescence (IF), Immunoblotting (IB) and Proximity Ligation Assays (PLA) are listed in the Reagent and Tools table.

For R-loop detection the S9.6 RNA:DNA hybrid-recognizing antibody was purified from the HB-8730 mouse hybridoma cell line

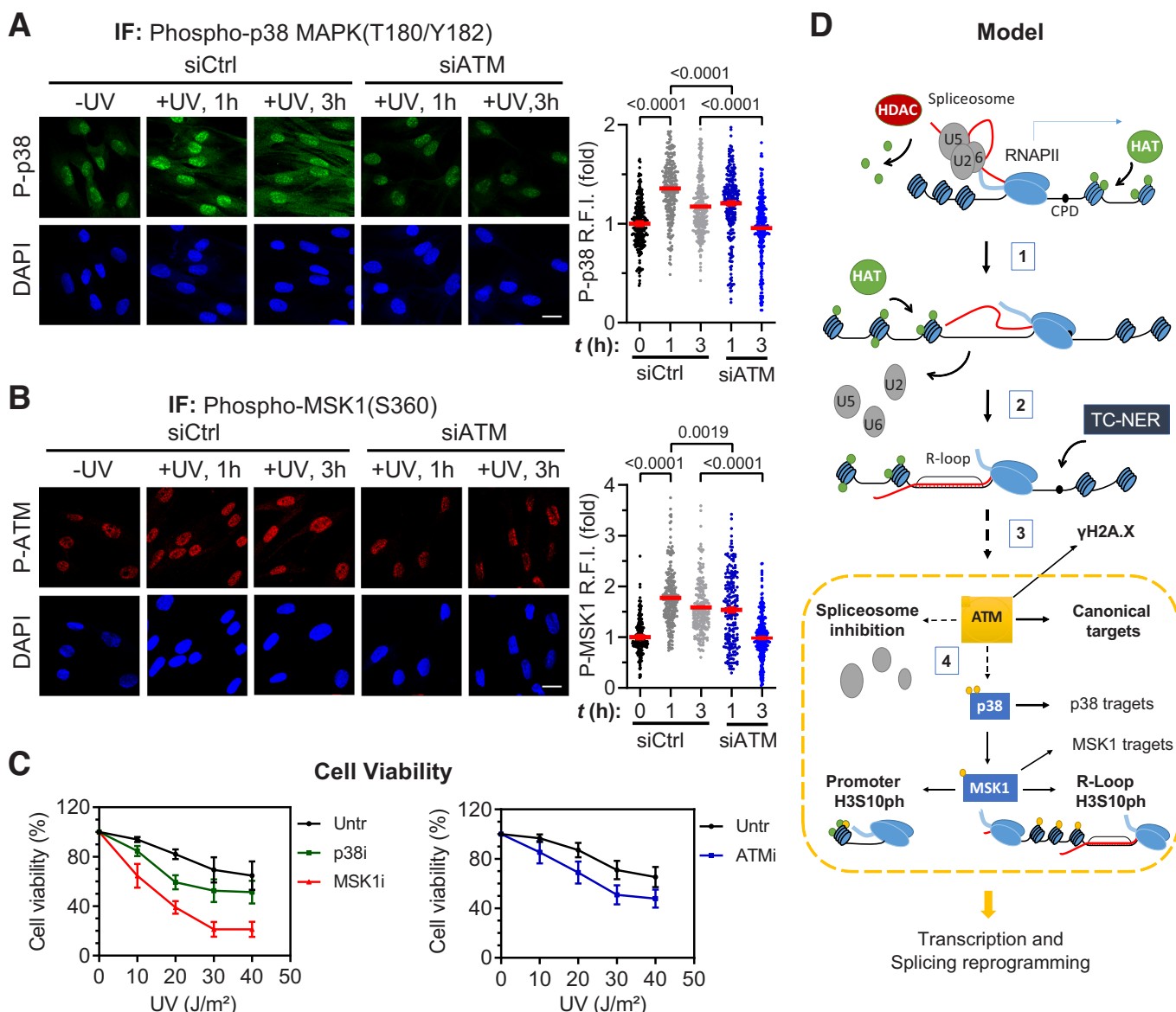

**Figure 7. ATM regulates the UV-induced activation of the p38/MSK1 signaling pathway.**

(A, B) ATM promotes UV-induced activation of the stress kinases p38 and MSK1. Representative immunofluorescence images and quantification of phosphorylated (activated) p38 (A) and its downstream effector MSK1 (B) in control or ATM-depleted quiescent HDFs following UV irradiation (40 J/m²) and fixation at indicated time points. Fluorescence intensities are normalized to control siRNA-transfected, non-irradiated cells. Nuclei were stained with DAPI. Scale bars: 20 μm. (C) ATM, p38, and MSK1 contribute to cell survival after UV-irradiation. Quiescent HDFs were pre-treated with ATM inhibitor KU55933 (10 μM, 1 h), p38 inhibitor SB203580 (20 μM, 4 h), or MSK1 inhibitor SB747651A (25 μM, 4 h) prior to UV exposure at indicated doses. Cell viability was assessed 48 h post-irradiation using AlamarBlue™. Data represent mean fluorescence intensities relative to untreated, non-irradiated controls (± SEM), from four (ATM) or three (p38, MSK1) independent experiments performed in technical triplicates. (D) Model illustrating chromatin–ATM cross-regulation in response to transcription-blocking DNA lesions (TBLs). *Stage 1:* Stalling of elongating RNAPII at TBLs induces HAT redistribution and chromatin relaxation upstream of the stalled transcription complex. *Stage 2:* Chromatin opening destabilizes spliceosomes, promoting their release from nascent transcripts, enabling RNAPII backtracking and TC-NER initiation. *Stage 3:* Free, unspliced RNA invades the relaxed DNA helix forming R-loops, which trigger ATM activation. *Stage 4:* ATM orchestrates genome-wide chromatin responses by phosphorylating downstream targets directly (e.g., H2A.X) and indirectly via the p38/MSK1 axis, which drives H3S10 phosphorylation. H3S10ph coordinates multiple chromatin events, including compaction of R-loop-rich regions and activation of stress response genes. Data information: Mean ± SEM of cells analyzed (left to right): (A) n = 264/253/276/254/256 and (B) n = 216/222/206/194/276; all from three biological replicates. Brown–Forsythe and Welch's ANOVA with Games–Howell's multiple comparisons test. Source numerical data, detailed statistical analysis, and unprocessed images are provided in the Source data file. Source data are available online for this figure.

(ATCC). Cells cultured in DMEM/10% FBS were adapted to PFHM-II serum-free growth medium suitable for MAb production (Gibco, Life Sciences) as recommended by the supplier. S9.6 monoclonal antibodies (mouse IgGs) were purified from 0.45 micron-filtered hybridoma supernatant by column chromatography using a HiTrap™ MabSelect SuRe™ column on an AKTA START protein purification system equipped with an automated fraction collector (Cytiva) as recommended by the manufacturer. Eluted antibody purity was verified by SDS-PAGE followed by Collodial Coomassie blue R-250 staining. CSB–mScarlet-I immunoprecipitations were performed with ChromoTek RFP-Trap® Agarose Beads.

### RNA interference

For gene silencing siRNA transfections were performed using Lipofectamine RNAiMAX (Invitrogen), according to the manufacturer's instruction with minor modifications. Briefly, cells were transfected 24 h after seeding in six-well plates (~70–80% confluence) by addition of 4 μL RNAiMAX and 4 μL 20 μM siRNA diluted in 150 μL optiMEM, in 1.5 ml F10/15% FBS. Following a 6–8 h incubation period, cells were re-transfected with freshly prepared complexes in low serum media (F10/0.5% FBS) which was replaced with serum-free F10 growth medium the next day. Experiments were performed ~48 h later. When two siRNA were used for double depletions, transfection complexes were prepared with 6 μL RNAiMAX and 3 μL of each siRNA.

### UV-C laser irradiation

For DNA damage infliction cells were irradiated using a germicidal UVC lamp (254 nm; TUV lamp, Phillips). For local UVC damage infliction, cells grown on glass coverslips were UV-irradiated through isopore polycarbonate membranes. Radiation doses and post-irradiation incubation times are stated in the figure legends.

### Fluorescence confocal microscopy (IF, RNA synthesis, PLA)

Changes in protein abundance or PTM levels were evaluated by immunofluorescence-detection using quiescent human dermal fibroblasts (HDFs) grown on glass coverslips which were fixed in 3.7% formaldehyde/0.15% Triton X-100/PBS for 10 min and further permeabilized with 0.5% Triton X-100/PBS. For CPD immunodetection, cells were pre-extracted with 0.1% Triton X-100/PBS (0.5 min, RT) and nuclear DNA was denatured with 0.07 N NaOH/PBS (5 min, RT) after fixation. Non-specific antigens were blocked in 3% bovine serum albumin (BSA)/PBS for 1 h. For S9.6 immunofluorescence, cells were pre-extracted with ice-cold 0.5% NP-40/PBS for 3 min followed by an 1 min extraction with 0.25% Triton X-100/PBS. Subsequently, cells were fixed in 100% Methanol (−20 °C, 10 min), acetone-permeabilized (−20 °C, 1 min) and blocked in 3% BSA, 0.1% Triton X-100/ 4×SSC. In all IF experiments incubation with primary antibodies was either in 3% BSA/PBS, or in 3% BSA /4×SSC (S9.6 ab) overnight at 4 °C, in a humidified chamber, and with Alexa Fluor™-conjugated secondary antibodies in 3%BSA/1 μM DAPI/PBS for 1 h at RT. Coverslips were mounted on glass slides using Aqua-Poly/Mount and fluorescence signals were visualized using a Zeiss LSM700 upright laser-scanning confocal microscope.

Relative changes in RNA synthesis were determined by pulse ethynyluridine (EU)-labelling followed by Click-iT conjugation of Atto 594 Azide. Serum-deprived HDFs grown on coverslips, were chemically treated and/or UV-irradiated and subsequently incubated with EU (300 μM, 1 h) which was added in the serum-free culture medium. Following fixation (3.7% formaldehyde, 15 min), permeabilization (0.1% Triton X-100/PBS,10 min) and blocking (1.5% BSA/PBS, 10 min), cells were incubated with the alkyne-azide. Following addition of Clickit chemistry reaction mix (50 mM Tris buffer pH 7.6; 60 μM Atto 594 azide; 4 mM CuSO$_4$*5H$_2$O; 10 mM freshly prepared ascorbic acid) cells were incubated for 1 h at RT in the dark. Nuclei were stained with DAPI/PBS for 10 min and coverslips were mounted using Aqua-Poly/Mount.

Constitutive and induced protein co-localization or their proximity to DNA damage sites (CPDs) or R-loops, was evaluated by Proximity Ligation Assays (PLA) performed with serum-deprived HDFs seeded on glass coverslips that were mock-treated or UV irradiated with 30 J/cm². Cells were fixed, blocked and incubated with each set of primary antibodies raised in different species (Mouse/Rabbit) using IF protocols described above. PLAs were performed using the Duolink® Proximity ligation assay (PLA®) Kit according to the supplied protocol which included a Duolink® PLA probe-set incubation, a ligation and an amplification step. Cells were mounted on glass slides using the provided Duolink® In Situ Mounting Medium with DAPI.

For all fluorescence microscopy experiments, image acquisition was performed using a Zeiss LSM700 upright laser-scanning confocal microscope equipped with a ×40 Plan-apochromat 1.3 NA or ×63 Plan-apochromat 1.4 NA oil immersion lenses (Carl Zeiss Micro Imaging Inc.). Fluorescence signal intensities were quantified using the ImageJ software (NIH) and dedicated macros developed in the ErasmusMC Optical Imaging Centre. In each experiment >100 cells per condition were analysed and all experiments were performed three times (biological replicates) unless otherwise stated.

### Live cell confocal microscopy (FRAP, UVC-laser irradiation)

Live cell imaging experiments were performed with a Leica TCS SP5 AOBS laser-scanning confocal microscope equipped with an environmental chamber (37 °C, 5% CO$_2$). For kinetic studies of GFP-tagged proteins, DNA damage was inflicted locally (in an approximately 1 nm subnuclear area) using UVC(266 nm) -laser micro-irradiation as previously described (Tresini et al, 2015). Briefly, a 2-mW pulsed (7.8 kHz) diode-pumped, solid-state laser emitting at 266 nm (Rapp OptoElectronic) was connected to the confocal microscope with an Axiovert 200 M housing adapted for UV by All-Quartz Optics. By focusing the UVC-laser inside nuclei without scanning, only a limited area within the nucleus (diffraction-limited spot) was irradiated. Cells seeded on quartz coverslips, were imaged and irradiated through an Ultrafluar quartz ×100, 1.35 NA glycerol immersion lens (Carl Zeiss Micro Imaging Inc.). Image analysis was performed using the LASAF software (Leica). For each image, fluorescence intensity in the irradiated area or a non-irradiated area in the nucleus (used to monitor possible photobleaching) were normalized to pre-irradiation levels in the same area. Data were expressed as fold change in fluorescence intensity.

Mobility of GFP-tagged proteins was measured by strip-FRAP using a ×63/1.4 NA HCL PL APO CS oil-immersion lens. Briefly, a narrow (~1 μm) strip spanning the width of the nucleus was photobleached at ≈20% of the initial GFP-signal intensity using a 488 nm-laser at 100% power. Recovery of fluorescence in the strip was

monitored at 22-ms intervals until a steady-state level was reached. For each cell, 200 pre-bleaching, 4 during bleaching and 1000 post-bleaching, images were obtained and quantified using the LASAF software. Fluorescence recovery was determined by subtracting the background fluorescence signal (measured outside the nucleus) from the nuclear fluorescence intensity followed by normalization to "pre-bleach" fluorescence intensity.

### Detection of histone post-translational modifications

Serum-deprived, confluent VH10T HDFs were mock-treated or irradiated with 40 J/m$^2$ UVC. Histones were extracted 4 h post-irradiation using a Histone Extraction Kit (Abcam) according to the manufacturer's instructions, and quantified against a BSA standard curve using the Pierce™ BCA Protein Assay Kit. Histone H3 and H4 modifications were detected simultaneously using the Histone H3 and H4 Modification Multiplex Assay Kits (Abcam) according to the provided protocols, using 70 ng histone extract per well for Histone H3, or 500 ng for Histone H4. Absorbance was measured at 450 nm and 655 nm (reference wavelength) in a Promega Glomax® Multimode reader. Optical Density (OD) values at 450 nm were corrected by subtracting the 655 nm OD and subsequently, average values of duplicate samples for each PTM were normalized to total histone H3 or histone H4 levels.

### Whole cell lysates and chromatin fractionation

All samples were prepared at 4 °C, unless otherwise specified. In all sample preparation and immunoprecipitation protocols, buffers were supplemented with the following inhibitors: 1×cOmplete™ EDTA-free protease inhibitor cocktail (Roche), 1× Phosphatase inhibitor cocktail 2 (Sigma), 1× Phosphatase inhibitor cocktail 3 (Sigma), 5 mM Na$_4$P$_2$O$_7$, 10 µM TSA, 15 µM MG132 and 1 mM PMSF. Sample protein levels were quantified using the Pierce™ BCA Protein Assay Kit.

For whole cell lysate preparation, cells were harvested/lysed in RIPA buffer [50 mM Tris pH 7.5, 150 mM NaCl, 2 mM MgCl$_2$, 0.1% SDS, 0.5% sodium deoxycholate, 1% NP-40]. Following benzonase digestion of nucleic acids (4 °C, 30 min), EDTA was added to 5 mM final concentration and lysates were clarified by centrifugation (16,000 rpm, 30 min). Supernatants containing solubilized proteins were diluted with 4× reducing Laemmli sample buffer (240 mM Tris pH 6.8, 8% SDS, 40% glycerol, 50 mM EDTA, 1 M DTT, 0.04% bromophenol blue) and processed for SDS-Polyacrylamide Gel Electrophoresis (PAGE).

Native chromatin was isolated after Triton-X 100 extraction and MNase digestion as previously described (Tresini et al, 2015). Briefly, cells were harvested in resuspension buffer [10 mM PIPES pH 7.0, 3 mM MgCl$_2$, 100 mM NaCl, 300 mM Sucrose] and soluble proteins were extracted with the addition of Triton-X 100 at 0.5% v/v final concentration. Following centrifugation (650× g, 5 min), depleted nuclei were washed twice in MNase digestion buffer [50 mM Tris pH 7.5, 4 mM MgCl$_2$, 50 mM KCl, 300 mM Sucrose] and subsequently were incubated with 0.3 U MNase per $1 \times 10^6$ nuclei, and 1 mM CaCl$_2$ (37 °C, 10 biological replicates). Addition of (NH$_4$)$_2$SO$_4$ to a final concentration of 250 mM was used to facilitate extraction of stably DNA-bound proteins. EGTA and EDTA were added to 5 mM final concentration and samples were centrifuged at 16,000× g for 20 min. Supernatants containing digested chromatin were processed for SDS-PAGE.

### Immunoprecipitations (ChIP, DRIP)

Native chromatin used for DRIP experiments was prepared by a 30-min Triton-X 100 extraction of cells harvested by trypsinization in lysis/IP buffer [30 mM HEPES pH 7.6, 130 mM NaCl, 2 mM MgCl$_2$, 0.5% Triton X-100]. Pelleted nuclei were washed twice with the same buffer, collected by centrifugation (750 × g) and subsequently sonicated using a Bioruptor Sonicator (Diagenode) (10 cycles of 15 s ON/45 s OFF) at the "high" setting. Clarified lysates containing equal amounts of protein were used for S9.6 –DRIP.

For DRIP experiments, the S9.6 antibody or the IgG control were pre-bound to BSA/ssDNA-blocked protein A agarose beads in 50 mM HEPES /KOH pH 7.5, 0.14 M NaCl, 5 mM EDTA, 1% Triton X-100, 0.1% Na-Deoxycholate and 1×1×cOmplete™ protease inhibitor cocktail, for 4 h with rotation. Immunoprecipitations were carried out O/N on a rotating wheel. Bead-bound complexes were collected by centrifugation, washed five times with ice-cold IP buffer and bound proteins were eluted by incubation at 95 °C for 10 min in 2× Laemmli buffer, prior to SDS-PAGE.

Digestion of cross-linked chromatin to nucleosomes, was performed by a modified protocol based on that of Pchelintsev et al (Pchelintsev et al, 2016). Briefly, cells harvested by trypsinization were cross-linked with 1% formaldehyde/PBS (RT, 8 min) on a rotating wheel. Formaldehyde was quenched by addition of glycine to 0.125 M final concentration. Cells collected by centrifugation (300 × g) were washed twice with ice cold PBS, re-suspended ($20 \times 10^6$/ml) in extraction/lysis buffer [50 mM Tris pH 8.0, 50 mM NaCl, 1% Triton X, 0.1% SDS, 1 mM EDTA pH 8.0] and extracted twice for 15 min each time, on a rotating wheel. Samples were centrifuged (750 × g), the pellet was washed with sonication buffer [5 mM Tris pH 8.0, 5 mM NaCl, 0.1% SDS, 0.05% Triton] and subsequently resuspended in the same buffer at $\sim$15 × 10$^6$ cells/ml. Samples were sonicated in 300 µl aliquots using a Bioruptor Sonicator (Diagenode) (10 cycles of 15 s. ON /45 sec. OFF) at the low setting. For benzonase digestion, the composition of sonicated chromatin samples was adjusted by the addition of 30 µl 10X Tris-Buffered Saline, pH 8.0, 30 µl 10% Triton-X 100, 5 µl 0.1 M MgCl$_2$ and 50 units Benzonase. Digestions, carried-out at RT for 15 min, were stopped by the addition of EDTA to 10 mM final concentration. Digested chromatin was used either for SDS-PAGE of in immunoprecipitation experiments.

Cross-linked immunoprecipitations were performed with equal amounts of crosslinked chromatin which was incubated with pre-blocked (0.5% BSA/200 ng/ml ssDNA, 4 h) RFP-TRAP Agarose or, with agarose binding control beads (ChromoTek). Following a 2 h-incubation with end-to-end rotation, beads were collected by centrifugation, washed five times with ice-cold IP buffer and bound proteins were de-crosslinked and eluted by incubation at 95 °C for 20 min in 2× Laemmli sample buffer (Sigma) prior to SDS-PAGE.

### Immunoblotting

Equal amounts of protein from each sample were size-fractionated on 5–20% gradient SDS-polyacrylamide gels (BioRad) and electro-transferred onto either Nitrocellulose (0.45 µM Protran®) or PVDF membranes (0.45 µM; Immobilon® -P) using a BioRad Mini-Protean electrophoresis system. Abundance of proteins of interest was assayed using antibodies at concentrations recommended by their manufacturers. Binding of each primary antibody was performed overnight at 4 °C in Tween 20/Tris-buffered saline (20 mM Tris, pH 7.4, 150 mM

NaCl, 0.1% Tween 20) containing 5% w/v non-fat dry milk or, 5% BSA for Phospho-protein detection. Following binding of the appropriate anti-mouse or anti-rabbit Alexa fluorochrome-conjugated secondary antibody and extensive washing, proteins of interest were visualized using the Odyssey CLx Infrared Imaging System (LI-COR Biosciences) and software. Signal intensities in obtained images were quantified using the ImageQuant™ TL v8.0.0. analysis software.

### Digested DNA purification and Agarose gel electrophoresis

Sonicated and benzonase-digested cross-linked chromatin was diluted 1:10 with ddH$_2$O and combined with 2× reverse cross-linking buffer [100 mM Tris pH 8.0, 600 mM NaCl, 1.0% SDS, 100 mM EDTA pH 8.0] and 100 µg Proteinase K. Samples were incubated at 60 °C O/N, followed by the addition of 100 µg of RNAse A and further incubation at 37 °C for 1 h. DNA was purified with a PCR purification kit (Qiagen) according to the manufacturer's instructions. DNA samples were fractionated by electrophoresis onto a 1% Agarose/1× TBS gel containing 1 µg/ml Ethidium Bromide.

### Cell viability assays

Cells grown in 96-well plates were synchronized in quiescence by 72 h serum deprivation, treated in triplicates with chemical inhibitors for 4 h, washed with PBS and irradiated with the indicated UVC doses. Cell viability was assayed 72 h post-irradiation using the AlamarBlue™ Cell Viability Reagent (Invitrogen) according to the manufacturer's protocol. Fluorescence was measured 3 h after addition of the AlamarBlue reagent at 570 nm using a Promega Glomax® Multimode reader. After background subtraction, triplicate average values were normalized to those of untreated/non-irradiated cells.

### Data collection and analysis

Microscopy data were obtained using commercially available Leica LAS AF or Carl Zeiss LSM software, as indicted. Following immunoblotting proteins were visualized using the Odyssey® Imaging System. Data from BCA Protein Assays, Multiplex ELISA, and AlamarBlue™ Cell Viability assays were collected using a Promega Glomax® Multimode reader.

Data were analysed using the following software: Leica LAS AF and LAS X or Live Cell Imaging data analysis; Carl Zeiss LSM and ImageJ/Fiji software for immunofluorescence data analysis. Image Studio Lite was used for Western blot image acquisition. Signal intensities in obtained images were quantified using the Image-Quant™ TL analysis software. Data were plotted and statistical analysis was performed using Prism GraphPad.

### Statistical analysis

Statistical tests were selected based on data distribution and group structure. Data were assessed for normality using the Shapiro-Wilk test. Variance homogeneity was evaluated using Brown–Forsythe and Bartlett's tests. In cases where data were normally distributed with equal variance, a one-way ANOVA followed by Bonferroni's multiple comparisons test (comparisons among multiple groups) was used. If these assumptions were not satisfied (and in multigraph panels that included datasets with both equal and unequal variances), statistical significance was determined using Welch's *t* test or Brown–Forsythe and Welch's ANOVA followed by Games–Howell multiple comparisons tests, as implemented in GraphPad Prism. This approach was particularly appropriate for our experimental findings, as it does not assume equal variances or sample sizes. When indicated, we verified results with non-parametric tests (e.g., Kruskal–Wallis with Dunn's correction), which supported the same conclusions. Protein kinetics in UV-C laser-irradiated live cells were compared with two-way ANOVA. No statistical method was used to pre-determine sample size. Sample sizes are consistent with those generally used in the field and reflect prior experience with the applied techniques. Randomization and blinding were not applied, as the experimental assays are not subject to systematic variation that would necessitate these measures. Statistical details are indicated in the figures and figure legends. Detailed statistical analysis for each figure is also provided in the Source data files.

## Data availability

This study includes no data deposited in external repositories.

The source data of this paper are collected in the following database record: biostudies:S-SCDT-10_1038-S44318-025-00537-7.

## Peer review information

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

## Acknowledgements

We acknowledge Dr L. Mullenders for his intellectual input and constructive criticism of the manuscript, the Optical Imaging Centre of ErasmusMC for technical support, Dr M. van Toorn for technical advice and Drs M van Sluis and JA Marteijn for the RPE1 CSB–mScarlet-I knock-in cell line. This work was funded by the World-Wide Cancer Research (WWCR) Foundation Grant 18-0023, European Research Council (Advanced Grant 340988-ERC-ID), the Dutch Cancer Society (KWF Grant 10506) and the Gravitation program CancerGenomiCs.nl from the Netherlands Organization for Scientific Research.

## Author contributions

**Irene Salas-Armenteros**: Conceptualization; Data curation; Formal analysis; Validation; Investigation; Visualization; Methodology; Writing—original draft; Writing—review and editing. **Maarten Klunder**: Data curation; Validation; Investigation; Methodology; Writing—review and editing. **Wim Vermeulen**: Conceptualization; Resources; Supervision; Funding acquisition; Project administration; Writing—review and editing. **Maria Tresini**: Conceptualization; Resources; Data curation; Formal analysis; Supervision; Funding acquisition; Validation; Investigation; Visualization; Methodology; Writing—original draft; Project administration; Writing—review and editing.

Source data underlying figure panels in this paper may have individual authorship assigned. Where available, figure panel/source data authorship is listed in the following database record: biostudies:S-SCDT-10_1038-S44318-025-00537-7.

## Disclosure and competing interests statement

The authors declare no competing interests.

# Expanded View Figures

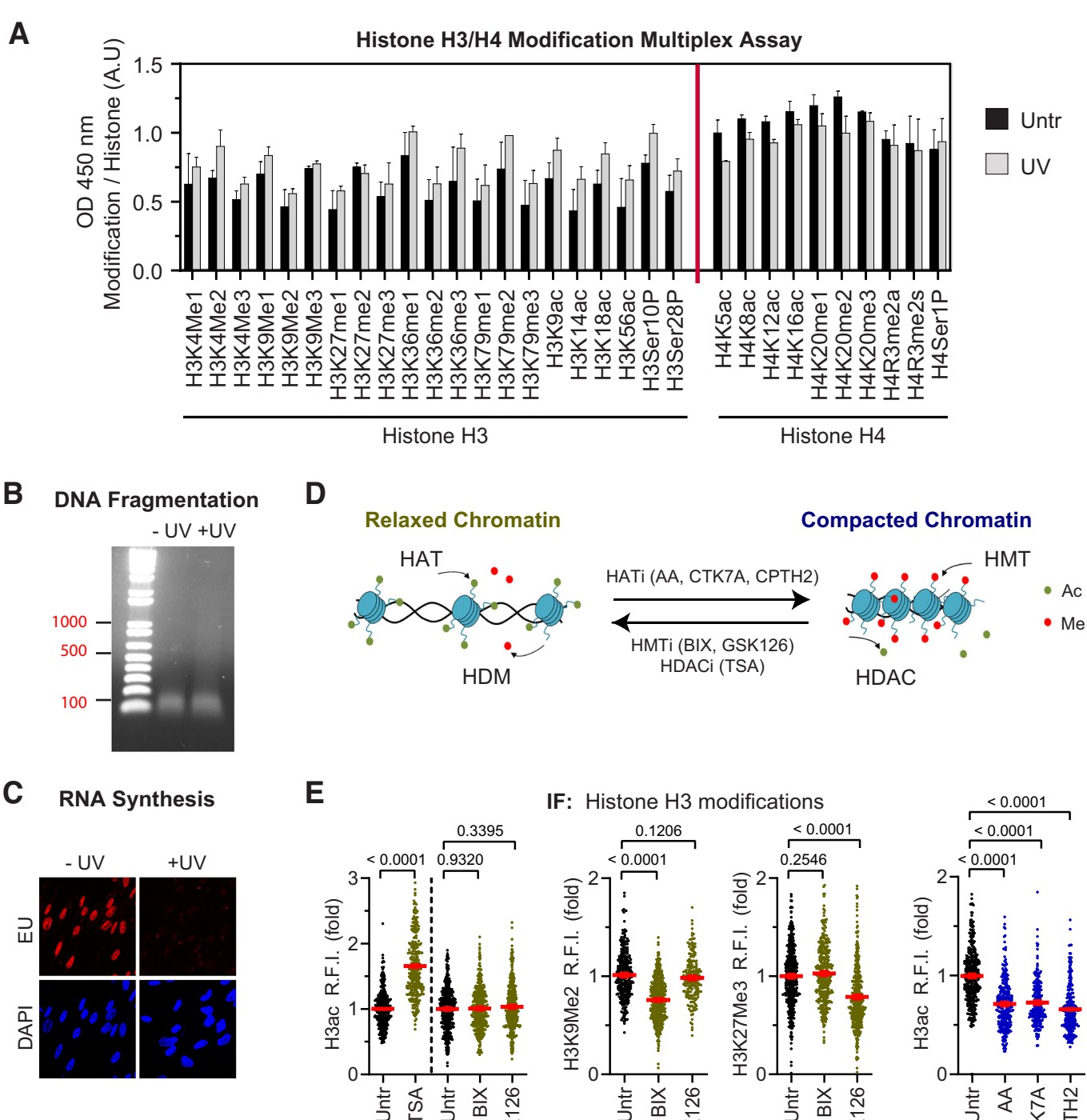

**Figure EV1. UV-induced chromatin remodeling and its pharmacological modulation.**

(A) Detection of UV-induced changes in histone H3 and H4 post-translational modifications (PTMs). Histones were extracted from quiescent HDFs, either untreated or UV-irradiated (40 J/m$^2$, 2 h). Thirty-one well-characterized histone H3 or H4 PTMs were measured using a multiplex ELISA-based colorimetric assay. Optical density (OD$_{450}$ nm) values for each modification were normalized to total histone H3 and H4 levels. Bars represent mean normalized OD$_{450}$ values from two independent biological replicates, each performed in duplicate. (B) Quality control of chromatin fragmentation used for chromatin immunoprecipitations (Fig. 1C). Agarose gel of DNA purified from cross-linked, digested chromatin is shown. DNA ladder: 1 Kb Plus (Invitrogen). (C) UV-induced transcription suppression. Representative immunofluorescence images of Ethynyl-uridine (EU) incorporation into nascent RNA in mock-treated or UV-irradiated (20 J/m$^2$, 2 h) quiescent HDFs. Nuclei were stained with DAPI. (D) Schematic illustrating the chromatin-modifying effects of pharmacological inhibitors used in this study. Chromatin relaxation is induced by the broad-spectrum histone deacetylase (HDAC) inhibitor Trichostatin A (TSA), which stimulates histone hyperacetylation, and by histone methyltransferase (HMT) inhibitors BIX01294 and GSK126, which prevent deposition of the suppressive marks H3K9me2 and H3K27me3, respectively. Chromatin compaction is promoted by histone acetyltransferase (HAT) inhibitors Anacardic Acid (AA), CTK7A (PCAF/p300 inhibitor), and CPTH2 (GCN5 inhibitor). (E) Effects of chromatin-modifying drugs on histone marks associated with relaxed or compacted chromatin. Quiescent HDFs were treated with the indicated inhibitors, followed by immunofluorescence detection of H3-specific PTMs in detergent-extracted nuclei. Acetylation changes were assessed using an antibody recognizing H3 acetylation at lysines 9, 14, 18, 23, and 27. H3K9me2 and H3K27me3 levels were detected following treatment with BIX01294 or GSK126, respectively. Nuclei were stained with DAPI. Plotted are mean fluorescence intensities normalized to untreated controls. Data information: (E) Mean ± SEM of (left to right): $n = 356/283/431/363/3949$ (three biological replicates), 124/98/93 (two replicates), 445/341/396 (three replicates), and 358/278/240/261 (three replicates). Unpaired two-tailed $t$-test with Welch's correction (H3ac by TSA), or Brown–Forsythe and Welch's ANOVA with Games–Howell's multiple comparisons test.

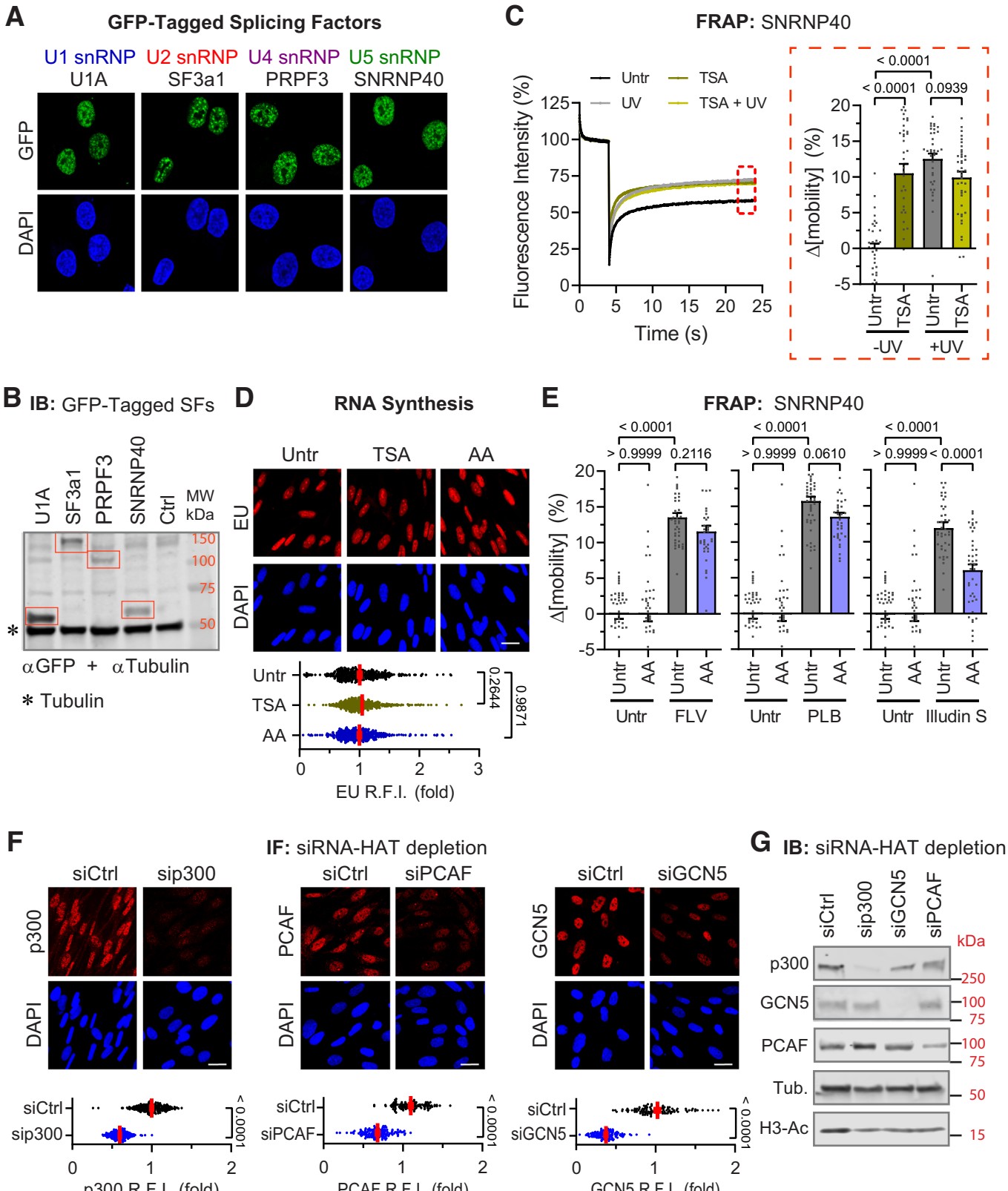

**Figure EV2.** **Validation of GFP-tagged spliceosomal proteins and histone acetyltransferase depletion efficiency.**

(A) Expression of GFP-tagged splicing factors (SFs) from distinct snRNP complexes in Human Dermal Fibroblasts (HDFs). Representative fluorescence microscopy images show characteristic speckled nuclear distribution of GFP-tagged SFs. Nuclei were stained with DAPI. (B) Immunoblot analysis of stably expressed snRNP-specific GFP-tagged SFs in HDFs. Tubulin served as a loading control. Depicted in each SNRNP40-GFP FRAP curve is the mean fluorescence recovery after background-correction and normalization to average pre-bleaching values ($n = 12$, representative experiment). Changes in mobility denoted as Δ[mobility] were calculated as the SNRNP40-GFP fluorescence in treated - fluorescence in untreated, non-irradiated cells at 20-21 s post photobleaching (indicated by the rectangle in the left panel) and plotted in the graph next to the FRAP curves. (C) SNRNP40-GFP mobility assayed by FRAP in quiescent HDFs under control conditions, following UV irradiation (20 J/m², 30 min), or chromatin hyperacetylation induced by TSA (1 μM, 3 h). Mean FRAP recovery curves are shown after background correction and normalization to average pre-bleaching values ($n = 12$, representative experiment). Mobility changes (Δ[mobility]) were calculated as the difference in normalized fluorescence intensity between UV irradiated and untreated non-irradiated cells at 20-21 s post photobleaching (indicated by the rectangle in the left panel) and plotted in the graph next to the FRAP curves.
(D) Chromatin acetylation does not impact global RNA synthesis. Representative immunofluorescence images and quantification of nascent RNA synthesis (EU incorporation) in quiescent HDFs treated with TSA (1 μM, 3 h) or Anacardic Acid (AA; 10 μM, 2 h). Mean fluorescence intensities were normalized to untreated controls. Nuclei were stained with DAPI. Scale bars: 20 μm. (E) HAT inhibition impairs spliceosome mobilization by transcription-blocking lesions (TBLs) but not by pharmacological disruption of transcription initiation or spliceosome assembly. SNRNP40-GFP FRAP was performed in quiescent HDFs treated with the transcription initiation inhibitor Flavopiridol (FLV - 1 μM, 1 h), the spliceosome maturation inhibitor Pladienolide B (PLB - 1 μM, 1 h), or the TBL inducing drug Illudin S (25 ng/ml, 1 h). Treatments were administered in the presence or absence of AA (10 μM, 2 h). Calculated changes in SNRNP40-GFP mobility are shown. (F, G) Efficiency of HAT depletion by siRNA. (F) Representative immunofluorescence images and quantifications showing decreased fluorescence intensities of p300, PCAF, and GCN5 following siRNA transfection. Nuclei stained with DAPI. Scale bars: 20 μm. (G) Immunoblots showing protein levels of p300, PCAF, GCN5, and Histone H3 acetylation in siRNA-transfected cells. Tubulin is shown as loading control. Data information: Mean ± SEM of: (C) $n = 34/37/39/42$ (left to right), (D) $n = 402/411/398$ (top to bottom), (E) $n = 40/10/15/35$, $40/40/37/38$, and $43/40/42/42$ (left to right); all from three (C, E) or four (D) biological replicates. (F) $n = 253/262$, $119/134$, and $107/138$ (left to right), from two biological replicates. (C–E) Brown–Forsythe and Welch's ANOVA, Games–Howell's multiple comparisons test; (F) unpaired two-tailed t-test with Welch's correction.

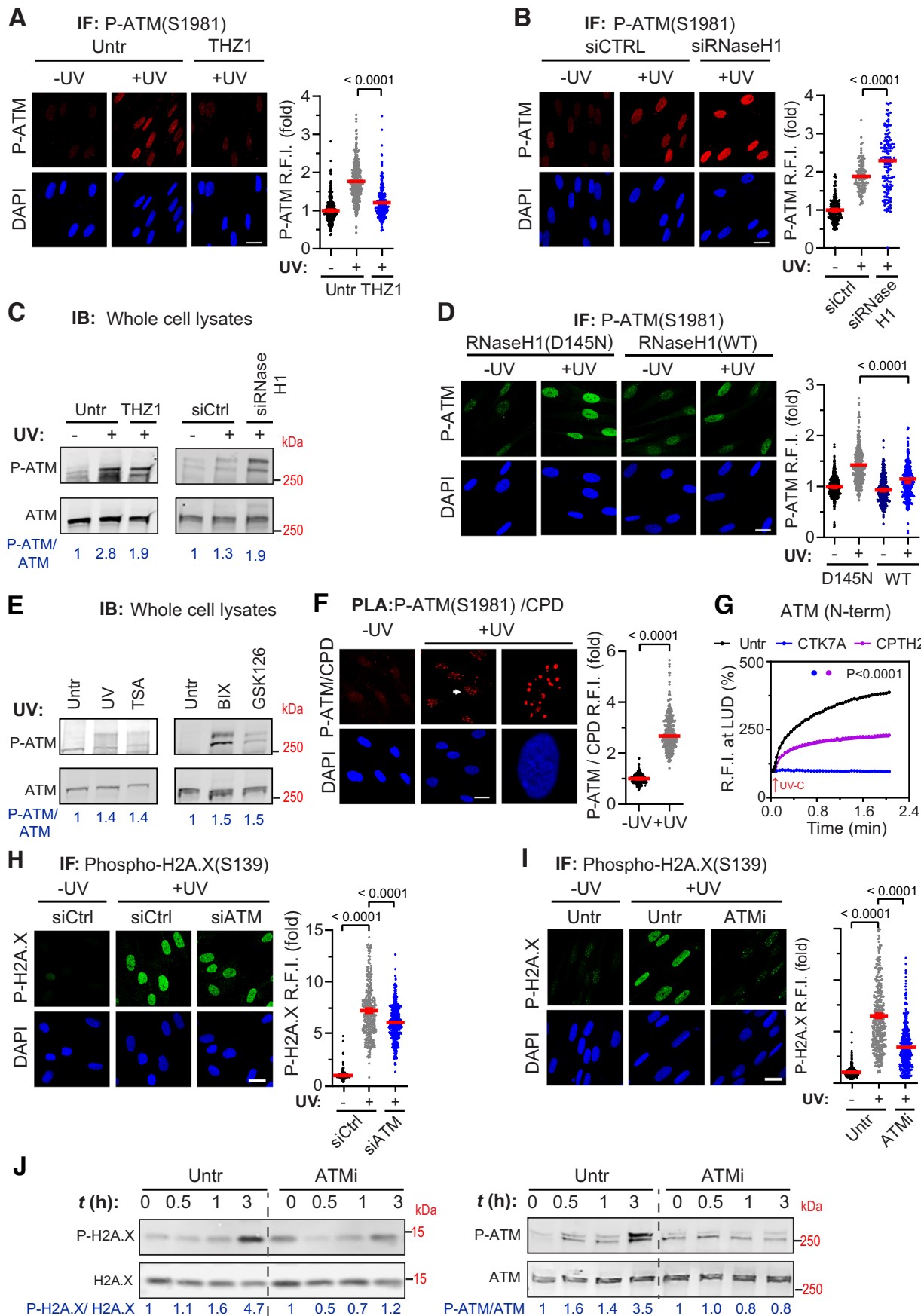

◀ **Figure EV3. Transcription- and R-loop-dependent ATM activation in response to UV-induced DNA damage.**

(A–D) UV-induced ATM activation requires active transcription and R-loop accumulation. Representative immunofluorescence images and quantification of UV-induced (40 J/m², 2 h) autophosphorylation of ATM in quiescent HDFs under the following conditions: (A, C) ± transcription inhibition by THZ1 (10 μM, 3 h); (B, C) following siRNA-mediated depletion of the R-loop hydrolase RNaseH1; (D) after doxycycline-induced expression of mCherry-tagged wild-type RNaseH1 or catalytically inactive D145N mutant. (E) Immunoblot analysis of ATM activation in response to chromatin-modifying treatments: UV irradiation (20 J/m², 2 h), HDAC inhibitor TSA (1 μM, 2 h), or HMT inhibitors BIX01294 and GSK126 (10 μM, 2 h each). All treatments induce chromatin acetylation typical of relaxed chromatin. (F) Representative images and quantification of Proximity Ligation Assays (PLA) detecting co-localization of auto-phosphorylated ATM (active) and UV-induced CPDs in quiescent HDFs. Arrow indicates magnified cell. (G) HAT inhibition impairs ATM recruitment to UV-C laser-damaged chromatin. Quiescent HDFs stably expressing GFP-tagged ATM N-terminal domain (aa 2–1314) were treated with HAT inhibitors prior to UV-C laser microirradiation (256 nm). Fluorescence intensities at the irradiation site were normalized to pre-irradiation levels. (H–J) ATM activity contributes to UV-induced phosphorylation of histone H2A.X. Representative images and quantifications of phospho-H2A.X in (H) siATM-depleted and (I) ATM-inhibitor treated cells. (J) Immunoblots showing phospho-H2A.X and ATM autophosphorylation at indicated times post-UV irradiation. Data information: Mean ± SEM of (left to right): (A) $n =$ 363/438/287; (B) 189/127/140; (D) 367/355/370/349; (F) 164/194; (H) 272/291/417; and (I) 338/443/393 cells; from three biological replicates (A, D, H, I) or two biological replicates (B, F). Brown–Forsythe and Welch's ANOVA with Games–Howell's multiple comparisons test (A, B, D, H, I) or unpaired two-tailed $t$-test with Welch's correction (F). (G) Mean ± SEM of $n =$ 41 (untreated), 43 (CTK7A-treated), and 35 (CPTH2-treated) cells from three biological replicates; repeated-measures two-way ANOVA.

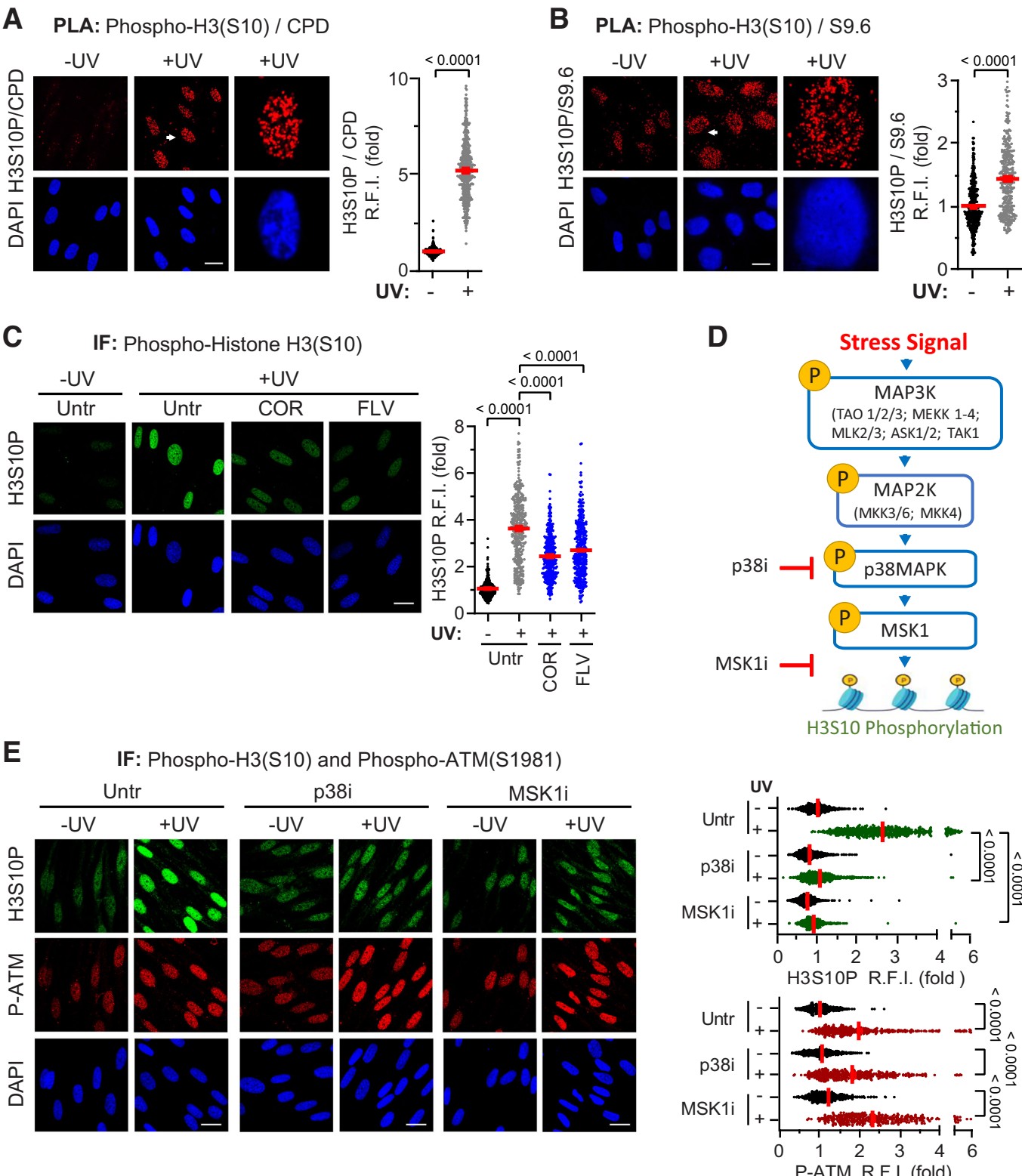

**A** PLA: Phospho-H3(S10) / CPD

**B** PLA: Phospho-H3(S10) / S9.6

**C** IF: Phospho-Histone H3(S10)

**D** Stress Signal

**E** IF: Phospho-H3(S10) and Phospho-ATM(S1981)

◄ **Figure EV4.   Transcription-dependent phosphorylation of Histone H3S10 at UV-damaged, R-loop-containing chromatin.**

(**A, B**) Phosphorylation of Histone occurs near UV-induced DNA damage and R-loop–rich chromatin. Representative PLA images and quantification of phosphorylation proximal to: (**A**) UV-induced CPDs and (**B**) RNA:DNA hybrids in detergent-extracted nuclei of quiescent UV irradiated (30 J/m², 1 h). Nuclei stained with DAPI. Scale bars: 20 µm. (**C**) Phosphorylation requires active transcription. Immunofluorescence images and quantification of H3S10P in quiescent HDFs treated with transcription inhibitors Flavopiridol (FLV, 1 µM, 4 h) or Cordycepin (COR, 10 µM, 4 h) prior to UV irradiation (40 J/m², 2 h). (**D**) Schematic of the p38–MSK1 signaling cascade mediating phosphorylation in response to stress, indicating pharmacological inhibitors used in this study. (**E**) UV-induced phosphorylation is dispensable for ATM activation. Representative images and quantification of P and auto-phosphorylated ATM in control and UV-irradiated quiescent HDFs with or without p38 (SB203580, 20 µM, 4 h) or MSK1 (SB747651A, 25 µM, 4 h) inhibition. Nuclei stained with DAPI. Scale bars: 20 µm. Data information: Mean ± SEM of (left to right): (**A**) $n = 569/484$; (**B**) 392/339; and (**C**) 351/348/357/374 cells from three biological replicates. Unpaired two-tailed *t* test with Welch's correction (**A, B**), or Brown–Forsythe and Welch's ANOVA with Games–Howell's multiple comparisons test (**C**). (**E**) Mean ± SEM of (top to bottom): $n = 495/532/539/540/477/479$ cells for both H3S10P and P-ATM detection from four biological replicates. Brown–Forsythe and Welch's ANOVA with Games–Howell's multiple comparisons test.

**A**

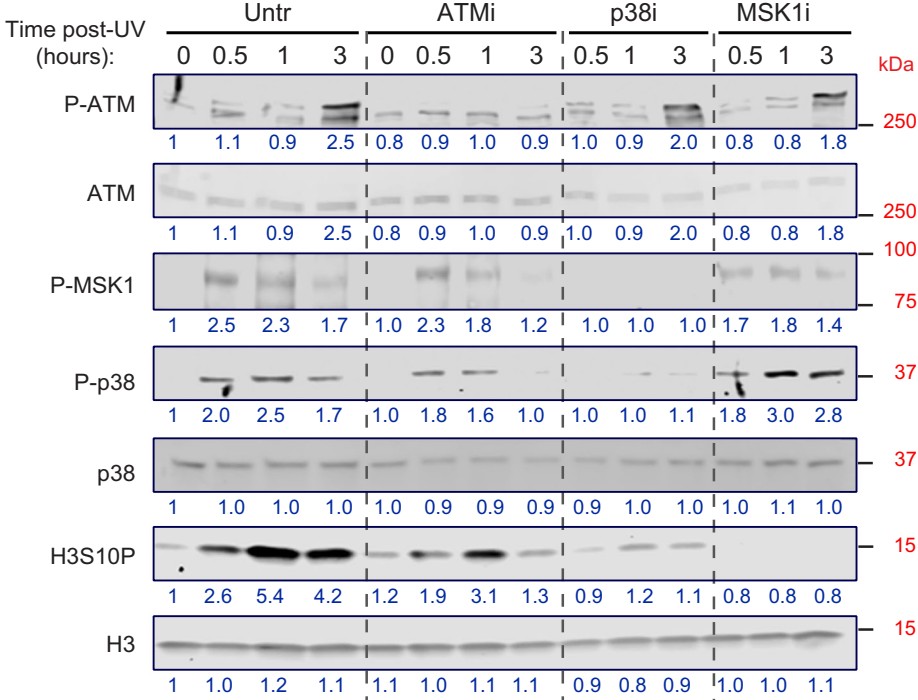

**B** IF: Phospho-H3(S10) and -ATM(S1891)

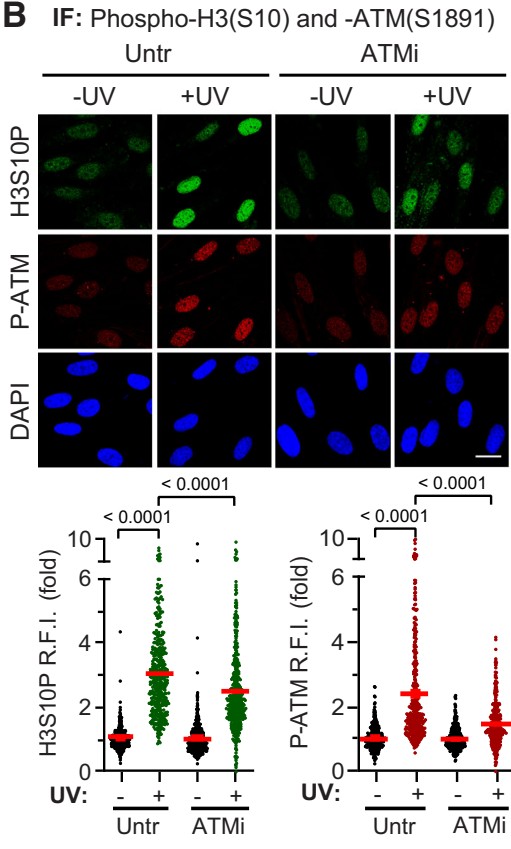

**C** IF: Phospho-p38(T180/Y182) and -MSK1(S360)

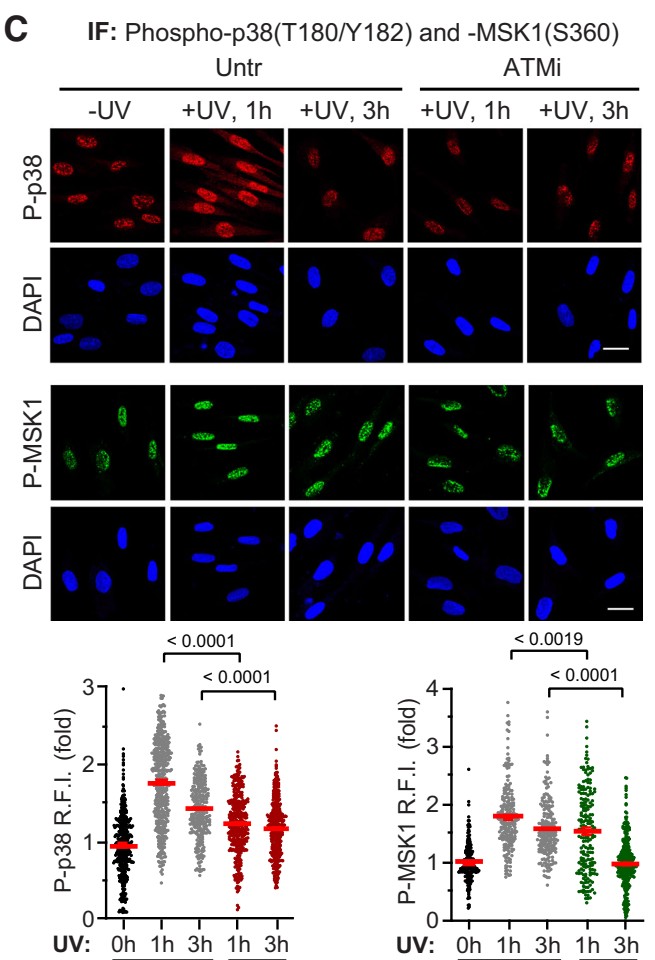

◀ **Figure EV5. ATM activity is required for UV-induced activation of the p38–MSK1- phosphorylation cascade.**

(**A**) Immunoblot validation of pharmacological inhibition of ATM, p38, and MSK1, and their effects on downstream targets regulating H3S10 phosphorylation. Quiescent HDFs were treated with KU-55933 (10 μM, 2 h), SB203580 (20 μM, 4 h), or SB747651A (25 μM, 4 h) prior to UV irradiation (40 J/m²). Whole cell extracts collected at indicated time-points post irradiation were analyzed by immunoblotting. All shown proteins were detected in the same membrane. (**B**) ATM activity promotes UV-induced phosphorylation of histone H3S10. Representative immunofluorescence images and quantification of phosphorylated H3S10 and auto-phosphorylated ATM in quiescent HDFs ± UV irradiation (40 J/m², 2 h) with or without ATM inhibitor KU-55933 (10 μM, 2 h) pre-treatment. (**C**) UV-induced activation of the p38/MSK1 pathway depends on ATM. Representative images and quantification of phosphorylated p38 and MSK1 with or without KU-55933 pre-treatment prior to UV irradiation (40 J/m², 2 h). Data information: Mean ± SEM of (left to right): (**B**) $n = 430/398/411/430$ (H3S10P and P-ATM); (**C**) 443/456/289/363/431 (P-p38); and 216/222/206/194/305 (P-MSK1) cells from three biological replicates. Brown–Forsythe and Welch's ANOVA with Games–Howell's multiple comparisons test.

