## [Peer Review File · The EMBO Journal]

Crosstalk between chromatin state and ATM signalling in DNA damage-induced transcription stress

Irene Salas-Armenteros, Maarten Klunder, Wim Vermeulen, and Maria Tresini

Corresponding author(s): Maria Tresini (m.tresini@erasmusmc.nl) , Wim Vermeulen (W.Vermeulen@erasmusmc.nl)

Review Timeline:

Submission Date:	21st Mar 25
Editorial Decision:	5th May 25
Revision Received:	10th Jul 25
Accepted:	29th Jul 25

Editor: Hartmut Vodermaier

Transaction Report:

This manuscript was transferred to The EMBO JOURNAL following peer review at another journal.

Reviewer #1

Overall, this is a good manuscript, despite being heavily reliant on the use of inhibitors, IF-based approaches and a unique cellular system.

Apart from that, the level of novelty and advancement in the field provided is not high enough, in my opinion, for publication. The role of spliceosome dissociation upon TBL is known as well as the R loops- and ATM-dependent mechanism at its basis. The role of chromatin relaxation and histone acetyl transferases in the DDR is also well established and studied, and confusedly analysed here. There is also very little new on ATM activation, H3S10P or stress-activated kinases.

Specific points:

1. The authors used many, non specific, inhibitors . The effects on spliceosome mobility seems “broad”, are observed in basal conditions (without UV treatment), using both HDAC and HMT inhibitors and upon silencing by siRNAs of basically all acetyl transferases. Many of these compounds (and knockdown) might be just affecting transcription or splicing indirectly; they also do it in the absence of DNA damage. I also find not clear/explained the involvement and role of Histone methylation. Finally, there are no direct proofs of different chromatin relaxation (or mobility) at sites of TBL.
2. why all of the three HATs are required for this process and how do they mechanistically act ? different genomic regions, specific histone modifications or effects on the spliceosome ?
3. The authors extensively used PLA, which is quite prone to artefacts. what about simply endogenous IFs or CSK chromatin extraction and WBs ?
4. TSA (so chromatin hyperacetylation) treatment causes ATM activation in the absence of TBLs. does this happen via increased R-loop formation ? this might argue for a disrupted transcription process or alternatively a chromatin relaxation mechanism independent from R loops.

Minor points

1. Why transcription inhibition does not block interaction between PCAF and GCN5 with CPD ? on the same line why there is no increased PLA signal in the PLA GCN5/S9.6.?

2. Effect of siRNAs should be measure by WB and not IF (HATs) and same applies to p ATM signalling
3. 4C, why WT RNaseH1 only modestly reduced H3S10 signal?
4. Signalling of ATM, MAPK, Histones etc. would be more clearly analysed by quantitative WB (at least in a few experiments)

Reviewer #2

This manuscript by Armenteros et al highlights the intricate cross-regulation between chromatin state and ATM signalling, which plays a pivotal role in the cellular response to transcription stress. In essence, the study is potentially intriguing for demonstrating how RNAPII stalling at TBLs initiates extensive chromatin reorganization and histone (hyper)-acetylation in proximity to the lesion, along with the involved regulatory networks. The study is well-designed and logistical conducted.

Specific points:

1. The study's reliance on microscope-based staining and live cell observation for assessing chromatin changes and spliceosome eviction may not provide a comprehensive picture of global chromatin alterations. Alternative methods, such as chromatin immunoprecipitation (ChIP) and combined sequencing techniques, could provide a more comprehensive understanding of dynamics of spliceosome eviction upon stress stimulation.
2. Is the eviction of the spliceosome from damaged chromatin specifically triggered by UV radiation? The author is advised to observe various types of genomic stress to investigate whether this behavior is unique or occurs under different conditions. To effectively examine this, Western Blot analysis would be an ideal approach to investigate dose- and time-dependent changes in these factors. A big gap in this study is that how does R-loop dependent ATM activation. This is a crucial aspect that requires clarification. Furthermore, the inclusion of the p38-MSK axis in the study remains puzzling, given that ATM can directly modulate the chromatin state and numerous protein kinases regulate H3S10 phosphorylation. Additionally, if this regulatory axis is indeed crucial for the genomic stress response, the authors should aim to identify the authentic phosphorylation sites of p38 mediated by ATM and thoroughly characterize their functional significance.

3. Immunostaining of p-ATM as a proxy for its kinase activity is not an optimal approach. Instead, the author is advised to employ western blot analysis or in vitro kinase activity assays to accurately assess ATM's kinase activity.

4. Could it be possible that H3S10 phosphorylation is directly influenced by altered chromatin states and H3K9 acetylation status, rather than through intermediary protein kinase regulatory networks?

5. The usage of different enzymes/kinase inhibitors should be examined to evaluate the inhibitory efficiency using specific target as readout.

Reviewer #3

In the manuscript titled “Cross talk between chromatin state and ATM signaling in DNA-damage induced transcription stress,” the authors demonstrate how transcription blocking lesion (TBL) activate a series of events, including spliceosome eviction, RNA: DNA hybrid formation, ATM activation, and chromatin modifications including H2AX and H3S10 phosphorylation in non-replicating human dermal fibroblast. Using FRAP and live cell imaging, the authors show that histone acetyl transferases play an important role in mature spliceosome eviction at transcription blocking lesions. Inhibition of histone acetyl transferases leads to suppression of ATM dependent R-loop formation. Moreover, author has shown that ATM activation impact chromatin modification directly by H2AX phosphorylation or indirectly H3S10 phosphorylation via p38/MSK1 signaling pathway. Manuscript is well described with detailed methodology. However, they are several concerns in the current versions.

Major issues:

1. It is hard to observe the changes in phosphorylation in representative images and this undermines confidence in the imaging quantitation. For example, Fig.3C sicontrol vs siPCAF and siCTRL vs sip300; Fig. 5A sicontrol +UV 1h vs siATM +UV 1h; Supplementary figure.8a; Fig.4a. Since these events are global phenomena, immunoblots would significantly enhance the manuscript.

2. The places DAPI panel (Fig. 3b group untreated (-UV) and Untreated (+UV); Fig. 4B +UV (FLV)) does not match the antibody group.

3. Does inhibition of ATM or silencing of ATM or UV treated for 3h impact on basal p38 expression ? In Fig. 5A and extended Fig. 10-b, it is important to include basal p38 expression data in each condition.

Minor issues:

1. Plotted data points are not visible mainly fig. 2a .

2. Missing y-axis label fig. 2b .

3. In most cases, author has used THZ1 to demonstrate transcription dependent phenotype except Fig. 4B. Why do they perform this assay using other inhibitors? Author is also advised to perform using THZ .

4. Page 8-line no.18-19 stating “UVC-induced histone PTMs in non-replicating HDFs, we observed a significant upregulation of H3S10P (extended fig.1b)”. In cited figure, there is roughly 15-20% difference between groups, and it is devoid of any statistical test. Therefore, it is hard to say significant differences. Please justify this sentence.

5. Page no. 9 line no. 8, there is wrong citation for figure. Fig.4A should be there instead of Fig.4C.

6. In Supplementary figure 2 legend, (F) and (G) is wrongly placed.

7. During transcription blocking events, histone acetylation transferases including p300, pCAF evicts spliceosome and forming the aberrant R-loop formation and ATM signaling events. Extended figure 7, shown that some ATM molecules are phosphorylated independently to R-loop, and relaxed histone induces pATM. Does it possible ATM directly impact on spliceosome eviction by changes on chromatin conformation?

RESPONSE TO REFEREES

We sincerely thank you for the opportunity to resubmit our manuscript titled "*Crosstalk between Chromatin State and ATM Signalling in DNA Damage-Induced Transcription Stress*" We appreciate the time and thoughtful consideration the editorial board and reviewers have dedicated to our work.

We thank the reviewers for their insightful suggestions to improve the quality of our manuscript. Their feedback helped us to clarify ambiguous points and prompted us to include additional data to substantiate our findings further.

In response to the reviewers' suggestions, we have revised the "Results" section to better highlight the rationale behind our approach, as well as the distinct aspects and significance of our findings. Additionally, we validated our initial results using diverse biochemical approaches, such as chromatin fractionation, immunoprecipitation, and immunoblotting. Detailed explanations are provided in the point-by-point response below. Here, we summarize the additional figures and findings incorporated into the revised manuscript.

1. **Figure 1b.** UV-induced acetylation of chromatin-associated Histone H3 (native chromatin fractionation/Immunoblotting)
2. **Figure 1c.** Chromatin acetylation of mono-nucleosomes associated with the TBL recognition-protein CSB. (Fragmentation of cross-linked chromatin to mono-nucleosomes/CSB Immunoprecipitation/Immunoblotting)
3. **Figure 3e.** Suppression of UV-induced ATM activation either by genetic or pharmacological inhibition of the Histone Acetyltransferases (HATs) p300, PCAF and GCN5. (Immunoblotting)
4. **Figure 4b.** Suppression of the UV-induced Histone H3(S10) phosphorylation by siRNA ATM silencing (Immunoblotting). Figure 4b also includes IBs showing the silencing efficiency of ATM siRNA.
5. **Figure 4c.** Suppression of UV-induced H3(S10) phosphorylation by the transcription inhibitor THZ1 (Immunofluorescence).
6. **Figure 4d.** Transcription dependency of UV-induced Histone H3(S10) phosphorylation. (Immunoblotting)
7. **Figure 4e.** Enhanced UV-induced phosphorylation of H3(S10) in response to RNaseH1 silencing. (Immunoblotting)
8. **Figure 4g.** Histone H3(S10) phosphorylation in proximity to R-loops partly depends on ATM activity (Chromatin isolation / S9.6 mab DNA:RNA Immunoprecipitation/Immunoblotting)

9. **Supplementary figure 1b.** Digestion of cross-linked chromatin to mono-nucleosomes by sonication/benzonase treatment. (DNA purification/ agarose gel electrophoresis)
10. **Supplementary figure 1c.** Inhibition of RNA synthesis (RNPII transcription) by UV irradiation. (EU incorporation/ "Click-it" chemistry/ Fluorescence Microscopy)
11. **Supplementary figure 3b.** Silencing efficiency of p300, PCAF and GCN5 siRNAs and their influence on H3Ac levels. (Immunoblotting)
12. **Supplementary figure 4c.** Increased HAT-chromatin association by UV-irradiation and its dependency on transcription. (Fractionation of cross-linked chromatin/Immunoblotting)
13. **Supplementary figure 7c:** UV-induced ATM phosphorylation depends on active transcription and R-loop levels. (Immunoblotting)
14. **Supplementary figure 7g:** ATM activation by chromatin relaxation induced by the Histone de-acetylation inhibitor TSA and the methyltransferase inhibitors BIX and GSK126. (Immunoblotting)
15. **Supplementary figure 8c:** Suppressed UV-induced H2A.X phosphorylation by pharmacological ATM inhibition (Immunoblotting). Figure also shows the inhibitor's efficiency in suppressing ATM auto-phosphorylation (Immunoblotting).
16. **Supplementary figure 10c.** Time course experiments showing the efficiency of pharmacological inhibition of:
 - a) MSK1 to completely suppress the UV-induced phosphorylation of Histone H3(S10). (Immunoblotting)
 - b) p38 to suppress the UV-induced MSK1 activating-phosphorylation and subsequently phosphorylation of Histone H3(S10). (Immunoblotting)
 - c) ATM to suppress its UV-induced autophosphorylation, and partly suppress the UV-induced p38, MSK1 and Histone H3(S10) phosphorylation.(Immunoblotting)
 - d) Lack of influence of ATM inhibition on p38 levels. (immunoblotting)
 - e) Lack of influence of the MSK1 and p38 inhibitors on ATM autophosphorylation (and levels). (Immunoblotting)

Please note that figure 10c depicts data from three combined experiments where all inhibitors were tested in parallel and protein levels were assayed on the same membrane.

POINT-BY-POINT RESPONSE

Reviewer 1:

“Overall, this is a good manuscript, despite being heavily reliant on the use of inhibitors, IF-based approaches and a unique cellular system”.

We appreciate the reviewer’s overall positive evaluation but we would like to comment on several aspects of the above statement.

We agree with the reviewer that combining the robustness of classical biochemical assays with the spatial resolution of our chosen methodologies would enhance the manuscript. To this end, we validated the majority of our findings using fractionations, immunoprecipitations, and immunoblotting, as detailed in our response to the reviewer’s specific comments (please see below).

We would like to emphasize that all conclusions drawn in this manuscript were based on findings using both pharmacological and, when applicable, genetic approaches. Examples of the latter were shown in the original manuscript in Figures 2c, 2d, 3c, 4a, 4c, 5a, 5b, and Supplementary Figures 3a, 7b, 7c, 8a). This was also stated on page 3 lines 21-22 of the original manuscript: *“The impact of chromatin structure on ATM activation by transcription stress was evaluated following pharmacological or genetic inhibition of histone modifiers prior to TBL induction”*. Similarly, genetic approaches were also used to address the role of ATM in influencing chromatin as depicted on the figures mentioned.

Importantly, we initially selected methods such as live-cell imaging, quantitative immunofluorescence, and proximity ligation assays because, although more labour-intensive than immunoblots, as they more accurately and quantitatively capture the responses of the spliceosome and ATM, as demonstrated in our previous work (Tresini, Nature, 2015). These approaches allowed us to study these intricate processes within the native cellular context, assess the heterogeneity of cellular responses, and analyse spliceosome dynamics. All figures depict the individual responses of hundreds of cells within the population.

Finally, we find the statement “unique cellular system” puzzling, as application of normal (diploid) human fibroblasts is rather common Biological research. For this reason, we can only surmise that the reviewers’ concern stems from our insufficient description of our chosen model. We amended the results section to list the specific cell lines used (initially mentioned only in the “methods” page

12 line 3 of the original manuscript) and elaborated on our initial explanation on the importance of using quiescent, non-transformed cells to conduct our studies. Both primary cell lines used (C5Ro established in Erasmus University Medical Centre and the ATCC-available VH10) have been extensively characterized in numerous publications in the literature.

These human diploid fibroblasts were particularly chosen to specifically capture the DDR to DNA damage-induced transcription-stress by eliminating replication-associated damage signalling, which would otherwise convolute data interpretation (see also Tresini, Nature, 2015).

“Apart from that, the level of novelty and advancement in the field provided is not high enough, in my opinion, for publication. The role of spliceosome dissociation upon TBL is known as well as the R loops- and ATM-dependent mechanism at its basis. The role of chromatin relaxation and histone acetyl transferases in the DDR is also well established and studied, and confusedly analysed here. There is also very little new on ATM activation, H3S10P or stress-activated kinases.”

Respectfully, we find the above statement unjustified for a number of reasons.

1. Our aim was not to establish a mechanistic link between spliceosome displacement from lesion-stalled RNAPII and ATM activation via the resulting R-loops, as this was indeed earlier described by us (Tresini, Nature, 2015). Instead, we sought to investigate how chromatin changes induced by transcription blocking lesions influence DDR signaling and the mechanisms underlying this process.
2. Extensive evidence links chromatin modifications to the cellular response to replication stress, double-strand DNA breaks, and lesions in non-transcribed regions, where chromatin decompaction is thought to facilitate repair system access. However, the influence of chromatin modifications on the response to transcription-blocking lesions (TBLs) remains insufficiently explored. This gap is largely due to the assumption that chromatin decompaction is unnecessary, as transcription typically occurs in already relaxed chromatin
3. The few published studies reporting chromatin alterations triggered by TBLs are focused on its impact on lesion repair (e.g. damage recognition, transcription recovery) rather than their influence on DDR signaling. In our previous work, we demonstrated that ATM activation by TBLs is triggered by spliceosome dissociation from lesion-arrested RNAPII immediately upon polymerase stalling, independent of TCR function. Furthermore, ATM activated by this mechanism has profound impact on spliceosome behavior and the cellular transcriptome. Building on this model, we investigated how chromatin modifiers influence this event and methodically examined their impact in every step of this ATM activation process.

4. We showed that TBL-activated ATM influences the chromatin landscape by 'translating' the initial acetylation signal into chromatin phosphorylation events, not only directly (histone H2.AX), but also indirectly (histone H3). Histone H3(S10) phosphorylation. This modification is linked to a wide range of cellular processes and can be catalyzed by a broad range of kinases activated by various signaling pathways in a cell cycle and stimulus dependent manner. Despite sporadic evidence from independent published studies that would support a link between ATM signaling and H3S10P (often with conflicting findings) a direct relationship between ATM signaling, MSK1 function and phosphorylation of H3(S10) has only been demonstrated in our current study.

Finally, we would like to emphasize that our observations demonstrating the extensive responses of non-replicating cells to pharmacologically induced changes in chromatin compaction, are likely to provide important and timely insights on the possible ramifications to post-mitotic tissues emerging from chromatin-modifying agents used for cancer management. These include the FDA-approved HDAC inhibitors Romidepsin (Istodax) and Vorinostat (Zolinza) and several HAT inhibitors currently in preclinical trials.

Specific points:

1. **“The authors used many, non specific, inhibitors . The effects on spliceosome mobility seems “broad”, are observed in basal conditions (without UV treatment), using both HDAC and HMT inhibitors and upon silencing by siRNAs of basically all acetyl transferases. Many of these compounds (and knockdown) might be just affecting transcription or splicing indirectly; they also do it in the absence of DNA damage. I also find not clear/explained the involvement and role of Histone methylation. Finally, there are no direct proofs of different chromatin relaxation (or mobility) at sites of TBL”.**

These are rather complex comments and we apologize in advance for our lengthy response.

First, we understand the potential for confusion as this is an extensive set of data involving numerous experiments requiring the use of induced chromatin perturbations with both, broad range and targeted inhibitors as well as genetic interventions.

It is important to note that only inhibitors that promote chromatin relaxation can influence spliceosome kinetics in non-damaged conditions. Inhibitors and siRNA mediated-HAT silencing that would promote chromatin compaction had no effect in unperturbed cells. So the interpretation that **“The effects on spliceosome mobility seems “broad”, are observed in basal conditions (without UV treatment).....and upon silencing by siRNAs of basically all acetyl transferases”** is not entirely correct.

“Many of these compounds (and knockdown) might be just affecting transcription or splicing indirectly.”

We are surprised by this comment, as we have clearly shown (and discussed) that global RNAPII transcription is not affected under the experimental conditions (*i.e.* low doses) applied to either stimulate or suppress spliceosome release from TBLs by chromatin modifiers. This was shown in Supplementary figure 2g of the original manuscript (now Supplementary figure 2f in the revised manuscript) and explained in text on page 5, lines 13-17 stating that: *“Notably, global RNA synthesis remained unchanged after incubation with AA (or TSA) with the applied doses, indicating that the spliceosome’s unresponsiveness to TBLs in the presence of AA, is not likely caused by reduced rate of lesion recognition by RNAPII in highly compacted chromatin.”* It should be pointed out that the observed levels of spliceosome mobilization would require substantial decrease in global RNA synthesis (above 90% inhibition) by transcription inhibitors as we have previously shown in dose response experiments (Tresini, Nature, 2015). Furthermore, changes in spliceosome mobility by transcription inhibition, or inhibition of spliceosome-maturation cannot be suppressed by HAT inhibitors, but only by the TBL-inducing agent Illudin S, supporting the specificity of chromatin compaction in suppressing spliceosome mobilization by TBLs (please see Supplementary fig.2f of the original manuscript, panel 2g in the revised manuscript).

“I also find not clear/explained the involvement and role of Histone methylation.”

We apologize for the complexity and concise descriptions. We found that chromatin relaxation (irrespective of whether histone acetylation is promoted or methylation is inhibited) stimulates spliceosome release. This indicates that the observed spliceosome mobilization is not simply the result of direct acetylation of either the splicing machinery or the transcription elongation complex. If that was the case, methyltransferase inhibition should have no impact on spliceosome’s behaviour. This was in fact the point of this experiment, to examine if it is the spliceosome or the transcription elongation complex that are being modified (as we initially suspected) or it relates directly to the chromatin state. These findings led us to the interpretation (page 5, lines 3-6 of the original manuscript) that: *“The PTM-dependent SF mobilization by these three functionally distinct inhibitors, suggests a process that is likely driven by chromatin relaxation rather than by specific modifications of chromatin-, transcription-, or splicing-associated proteins”*. We have not included further analysis on the effect of interfering with the methylation status of chromatin to avoid further complications.

“Finally, there are no direct proofs of different chromatin relaxation (or mobility) at sites of TBL.”

It is unclear to us what the reviewer would consider a direct evidence of chromatin relaxation at sites of TBLs, and how we should address this. It should be noted that TBL induction is stochastic, which makes it virtually impossible to 'directly prove chromatin relaxation at sites of TBL'.

In absence of a suggested approach, the only way we could envision to support that chromatin upstream of TBL-stalled RNAPII is relaxed by acetylation, was to immunoprecipitate CSB-associated mono-nucleosomes from UV-irradiated cells and show by immunoblotting that they are indeed acetylated (Figure 1c). For these experiments, isolated cross-linked chromatin was sonicated and benzonase-digested to fragments of approx. 150 nucleotides (verified by agarose gel electrophoresis shown in Supplementary figure 1b) using an optimized method similar to that of Pchelintsev et al (PMID: 26821228).

For proper interpretation of the data, it is important to consider the dynamics and binding position of CSB to TBL-stalled RNAPII (CSB is a crucial TC-NER initiating factor, required for TBL clearance). First, CSB transiently interacts with chromatin, and this interaction is stabilized when elongating RNAPII is arrested at a TBL (see, Llerena-Schiffmacher et al, PMID: 37716192). This is illustrated in Figure 1c showing the stabilized interaction between CSB and elongating RNAPII, recognized by the antibody targeting S2-phosphorylated CTD of RNAPII. Second, CSB binds to DNA upstream of TBL-stalled RNAPII, as demonstrated by structural studies (e.g. Kopic et al, PMID: 34526721). Therefore, it is unlikely that a CSB-associated mono-nucleosome would be located downstream of the stalled RNAPII. This suggests that the observed acetylated histone H3-containing nucleosomes are most likely upstream of the stalled polymerase. Please note that with unperturbed transcription chromatin is compacted upstream of elongating RNAPII. It is also important to note that RNAPII progression and chromatin relaxation are reciprocally coupled. In the absence of transcription, as with the UV-dose and time point post-irradiation used in our experiments (Supplementary figure 1c), there should be a substantial decrease in transcription-associated chromatin acetylation. In contrast, we observed TBL-induced H3-acetylation, which most likely occurred upstream of TBL-stalled RNAPII to allow spliceosome displacement, supporting our proposed model depicted in Figure 5d.

2. why all of the three HATs are required for this process and how do they mechanistically act ? different genomic regions, specific histone modifications or effects on the spliceosome ?

This is a rather convoluted and holistic question. Initially, the observation that inhibition of any of the three HATs (p300, GCN5 and PCAF) is sufficient to suppress spliceosome mobilization and all subsequent steps leading and including to ATM activation, was also puzzling to us.

Since inhibition of all three HATs suppresses global chromatin acetylation to similar levels (Supplementary figure 2b), and their inhibitory effect on spliceosome behaviour is synergistic

(Figures 2a, 2b, 2d) it would suggest that the level of inhibition is proportional to the extent of chromatin relaxation. This could be explained by either a redundancy of function or a compensatory mechanism, e.g. recruitment of the non-inhibited/non-depleted HATs to relax chromatin surrounding the abundant photolesions in non-transcribed regions facilitating lesion access and repair by GG-NER. Both redundant and compensatory HAT functions are supported by extensive literature showing both distinct and overlapping substrate specificity and function of these three HATs, their co-operative function in a stimulus and genomic location- dependent manner and their participation in complexes with similar function.

Acknowledging that a situation where chromatin processes are extensively disrupted by exogenous manipulations does not mirror of what happens in a physiological setting in response to DNA damage, we chose to ask the relevant question: Which HAT(s) acetylates chromatin proximal to TBLs. We reasoned that such a HAT should meet all of the following criteria: 1) Co-localize with CPDs (a profound TBL); 2) co-localize with R-loops that are formed as consequence of RNAPII TBL-stalling, and; 3) its localization should depend on ongoing transcription. As shown by PLA assays (figure 2e, Supplementary figure 4a, 4b, 5a and 5b), as well as by HAT recruitment to UV-irradiated subnuclear areas (Supplementary figure 4d) and the additional data provided by chromatin fractionation/immunoblotting (Supplementary figure 4c) the p300 HAT met all the above criteria (and to a much lesser extend PCAF).

Thus, we inferred that in a physiological setting, in response to CPD formation chromatin relaxation is achieved primarily by p300 when the lesion is detected by RNAPII and that GCN5 relaxes chromatin throughout the genome to allow lesion access by GG-NER.

3. The authors extensively used PLA, which is quite prone to artefacts. what about simply endogenous IFs or CSK chromatin extraction and WBs ?

First, please note that endogenous IFs detecting co-localization of HATs with CPDs were shown in Supplementary figure 4c of the original manuscript (current Supplementary figure 4d). Moreover, since chromatin association of HATs is dynamic, they do not remain chromatin-bound upon detergent extraction with CSK and are only detected in the nucleoplasmic- and not in the chromatin-fraction. However, upon the reviewer's request we developed a combined protocol using PFA cross-linking followed by extensive detergent extraction of non-chromatin bound proteins (1% SDS/1%Triton), chromatin digestion and immunoblotting. Typical results of these experiments are depicted in Supplementary figure 4c, confirming our PLA-findings.

However, we respectfully disagree with the statement that the PLA procedure we applied is prone to artefacts. We extensively optimized PLA's which are typically performed after chromatin cross-linking to accurately and reproducibly "capture" this transient interaction. Please note that all

experiments presented in this manuscript include positive and negative controls confirming the specificity of PLA signals.

4. TSA (so chromatin hyperacetylation) treatment causes ATM activation in the absence of TBLs. does this happen via increased R-loop formation ? this might argue for a disrupted transcription process or alternatively a chromatin relaxation mechanism independent from R loops.

The effect of TSA in promoting R-loop formation is well established in the literature and has also been confirmed by us (Salas-Armenteros, EMBO J, 2017). This supports our interpretation that ATM is activated by R-loops formed upon spliceosome displacement from transcription elongation complexes and is consistent with all the other data presented in the manuscript.

Minor points

1. Why transcription inhibition does not block interaction between PCAF and GCN5 with CPD ? on the same line why there is no increased PLA signal in the PLA GCN5/S9.6.?

As explained above, these observations point to a p300-specific role in chromatin acetylation when a CPD is detected by RNAPII, while PCAF and GCN5 likely relax CPD-containing chromatin throughout the genome to allow lesion access by GG-NER. We amended the text to further clarify this point.

2. Effect of siRNAs should be measure by WB and not IF (HATs) and same applies to p ATM signalling

As requested by the reviewer, we have confirmed our findings by immunoblotting. The new data are presented in Figure 4b (ATM silencing), Supplementary figure 3b (HAT silencing), in Figures 3e, 4b, and Supplementary figure 7c, 7g, 8c and 10c (ATM activation and signalling).

It should be noted that we do not fully comprehend why immunoblotting is preferred ('should be measured by WB') over IF, as both methods use the same antibodies to indirectly estimate cellular protein levels. In our humble opinion, IF has several advantages over immunoblotting as it is able to capture cell to cell variations and avoids the influence of non-specific enzymatic reactions resulting from cellular disruption.

3. 4C, why WT RNaseH1 only modestly reduced H3S10 signal?

RNaseH1-catalysed R-loop resolution is only one of the mechanisms employed by the cell to limit these structures, as it is becoming increasingly clear in the literature. Because RNaseH1 has additional cellular functions (e.g. Okazaki fragment resolution) generation of stably expressing

(active) enzyme is not possible due to toxicity. For this, others and we have developed inducible systems to transiently overexpress RNaseH1. For our experiments that was particularly important, as quiescent fibroblasts are impervious to transient transfection. To avoid artefacts either by the doxycycline used to drive expression, or by excessive RNaseH1 levels, we have optimized our inducible system to allow for an approximately 20-30% decrease in R-loops levels. This is consistent with the modestly reduced H3(S10) phosphorylation levels shown in the figure.

4. Signalling of ATM, MAPK, Histones etc. would be more clearly analysed by quantitative WB (at least in a few experiments)

We confirmed the vast majority of our data with Immunoblots as requested by the reviewer. These are illustrated in Figures 3e, 4b, 4d, 4e and Supplementary figure 7c, 7g, 8c and 10c. Please note that; to maintain the manuscript within reasonable size we included a combined time-course immunoblotting experiment in Supplementary figure 10c where we show the effects of ATM, p38 and MSK inhibitors on the phosphorylation and levels of phospho-ATM, ATM, phospho-MSK1, phospho-p38, p38, Histone H3, phospho-histone H3 (S10) and XAB2 (shown as a loading control). The time courses were performed in parallel and all protein abundances shown are on the same membrane.

Reviewer #2:

“This manuscript by Armenteros et al highlights the intricate cross-regulation between chromatin state and ATM signalling, which plays a pivotal role in the cellular response to transcription stress. In essence, the study is potentially intriguing for demonstrating how RNAPII stalling at TBLs initiates extensive chromatin reorganization and histone (hyper)-acetylation in proximity to the lesion, along with the involved regulatory networks. The study is well-designed and logistical conducted.”

We appreciate the reviewer’s positive comments and suggestions. We would like to comment on the statement: **“ However, there are numerous shortcomings and gaps throughout the manuscript. Most notably, the majority of the techniques employed in this study relies on microscope-based staining and live cell observation, which are artificial and cannot accurately represent global chromatin changes. These issues reduced the rigor of this study and thus overall diminishes enthusiasm for this work by this reviewer.”**

We agree with the reviewer on the clear advantages of combining biochemical assays, which reflect the response of the entire cell population, with live-cell imaging and immunofluorescence, which capture the responses of individual cells within the population. In the revised manuscript, we confirmed the vast majority of our findings using biochemical techniques, which we believe will enhance the reviewer’s enthusiasm for our work.

However, we are puzzled by the suggestion that the live-cell imaging approach could be considered artificial. On the contrary, we believe that studying protein behaviour in its natural context minimizes potential artefacts arising from the disruption of cellular structures. As noted in our manuscript (page 4 lines 4-6) and in our response to the first reviewer, we have conducted and previously published (Tresini et al, Nature, 2015) extensive biochemical experiments to validate the reproducibility, accuracy, and sensitivity of our methodology. These included chromatin fractionations and immunoblotting, immunoprecipitations of GFP tagged and endogenous splicing factors and elongating polymerase, chromatin isolation and PCR detection of spliceosome-associated snRNAs, splicing assays, and transcriptome sequencing for splicing changes.

Specific points:

- 1. The study's reliance on microscope-based staining and live cell observation for assessing chromatin changes and spliceosome eviction may not provide a comprehensive picture of**

global chromatin alterations. Alternative methods, such as chromatin immunoprecipitation (ChIP) and combined sequencing techniques, could provide a more comprehensive understanding of dynamics of spliceosome eviction upon stress stimulation.

This is essentially the same comment as addressed above. In the revised manuscript (as well as in our previously published work), we have included a series of cell fractionation and immunoprecipitation experiments combined with immunoblotting (see also response to reviewer #1) to confirm our microscopy-based evidence. However, on the premise that DNA damage occurs stochastically throughout the genome and thus RNAPII stalling at TBLs, we are not sure what to expect from ChIP-sequencing approaches and how sequencing would be informative on the behaviour of the spliceosome. Of note, we have previously applied -omics approaches such as transcriptome sequencing, to show extensive DNA damage-induced changes in splicing patterns and gene expression profiles that were dependent on ATM signalling. Those however, represented a secondary, organized response to the initial stochastic spliceosome release upon DNA damage.

2. Is the eviction of the spliceosome from damaged chromatin specifically triggered by UV radiation? The author is advised to observe various types of genomic stress to investigate whether this behavior is unique or occurs under different conditions.

This issue was already addressed in Tresini, et al, Nature, 2015. The initial spliceosome displacement is specifically triggered by TBLs either when induced by UV irradiation, or pharmacologically, by Illudin S. We have previously shown that this response is specific for this type of lesions and cannot be induced by other types of DNA damage that can be bypassed by RNAPII. To increase clarity, we amended the text in the manuscript to indicate this specificity (page 5, lines 3-5).

To effectively examine this, Western Blot analysis would be an ideal approach to investigate dose- and time-dependent changes in these factors.

Time-course and dose-response experiments combined with chromatin fractionations and immunoblotting have already been published in our previous manuscript (Tresini, et al, Nature, 2015). Importantly, those observations were also confirmed by live cell imaging approaches, which turned out to outperform fraction/immunoblotting procedures in terms of sensitivity and reproducibility. This is exactly the reason why we prefer to use the more quantitative, though tedious, FRAP procedure, as was also indicated on page 4, lines 4-10 of the original submitted manuscript (page 5, lines 1-9 in the revised manuscript).

“A big gap in this study is that how does R-loop dependent ATM activation. This is a crucial aspect that requires clarification”. The mechanism of ATM activation by R-loops is indeed a critical step of this pathway and we have extensively studied this event. In our previous work (Tresini, et al, Nature, 2015) we have provided complementary data showing that unlike ATM’s activation by dsDNA breaks neither dsDNA break formation nor the MRE11–RAD50–NBS1 (MRN) complex are required for ATM’s activation in response to R-loops. However, we respectfully disagree that this causes a big gap in the current study, as this manuscript is specifically focused on the interplay between chromatin dynamics, and ATM activation by TBLs. Of note, we are currently investigating which factors mediate ATM activation through its interaction with R-loops. However, we consider this to be beyond the scope of the present manuscript.

“Furthermore, the inclusion of the p38-MSK axis in the study remains puzzling, given that ATM can directly modulate the chromatin state and numerous protein kinases regulate H3S10 phosphorylation.”

We would like to elaborate on the reasoning behind the p38-MSK axis. The effect of various kinases in H3(S10) phosphorylation is both cell-cycle and stimulus dependent. In this manuscript we show that in quiescent cells photolesion-induced H3(S10) phosphorylation can be completely suppressed by inhibition of MSK1 and its upstream (within this context) activator, p38. This is illustrated in Supplementary figure 9d of the original manuscript (currently Supplementary figure 9e) and confirmed by immunoblotting as shown in Supplementary figure 10c of the revised manuscript. Although we inhibited most of the literature reported histone H3(S10) kinases (negative data were not shown), none other than the p38/MSK1 inhibitors was able to suppress its UV-induced phosphorylation in quiescent cells.

As briefly stated in the manuscript, our initial aim was to test whether Histone H3(S10) phosphorylation (H3S10P) marks, proximal to TBL-induced R-loops are involved in ATM activation. After identifying MSK1 as the kinase responsible for H3(S10) phosphorylation upon UVC irradiation of non-cycling cells (G0), we inhibited this reaction and assessed ATM activity, which surprisingly appeared unaffected. Thus, the opposite to our expectation was true: optimal H3(S10) phosphorylation actually required ATM activity. This led us to hypothesize that, after its activation, ATM may signal to promote compaction surrounding regions of R-loop/DNA damage-containing chromatin to insulate it from upstream transcription elongation complexes.

Further evidence supporting this hypothesis are illustrated in Figure 4g of the revised manuscript, where DRIP experiments demonstrated co-immunoprecipitation of S10-phosphorylated histone H3 and the TC-NER protein CSB (a marker of TBL-containing chromatin) with RNA:DNA hybrids in UV-

irradiated cells. The manuscript was accordingly amended to include interpretation of these findings (page 10, lines 17-20).

“Additionally, if this regulatory axis is indeed crucial for the genomic stress response, the authors should aim to identify the authentic phosphorylation sites of p38 mediated by ATM and thoroughly characterize their functional significance”.

Reduced p38 activity was deduced by the reduced phosphorylation of its downstream target MSK1 (Figure 5b) and by the reduced levels of p38’s dual phosphorylation of the Thr–Gly–Tyr motif (Thr180/Tyr182 of the p38a isoform, Figure 5a), which is necessary to fully activate the kinase. This dual phosphorylation is catalysed by dual specificity Ser/Thr, Tyr kinases, and its sequence does not conform to the SerGln/ThrGln (SQ/TQ) motif recognized by ATM which, as a single specificity Ser/Thr kinase is not able to phosphorylate Tyr residues.

For this, we inferred that ATM does not function on p38 directly, but to an upstream factor of p38 activation. Since ATM has been previously shown to phosphorylate the TAO kinase in response to ionizing radiation and its inhibition prevents this event as well as the downstream p38 activation, we inferred that TAO (or a related kinase) is the most likely target of ATM. Although there is an ATM recognition SQ/TQ motif in p38 at Thr560, which could potentially be phosphorylated by ATM, we think that investigating if and how this could affect p38’s Thr180/Tyr182 phosphorylation is beyond the scope of the current study.

3. Immunostaining of p-ATM as a proxy for its kinase activity is not an optimal approach. Instead, the author is advised to employ western blot analysis or in vitro kinase activity assays to accurately assess ATM's kinase activity.

In the revised manuscript we included Immunoblots showing both the abundance of auto-phosphorylated ATM (widely used in the literature to demonstrate ATMs activity) and the abundance of the phosphorylated form of its downstream target Histone H2A.X, confirming the immunostainings. These are illustrated in figure 3e and Supplementary figure 7c, 7g, 8c and 10c. Additionally, we included immunoblots showing the influence of ATM silencing and inhibition on H3(S10) phosphorylation (Figure 4b and Supplementary figure 10c).

4. Could it be possible that H3S10 phosphorylation is directly influenced by altered chromatin states and H3K9 acetylation status, rather than through intermediary protein kinase regulatory networks?

Although we cannot exclude an influence of histone H3 acetylation status on its phosphorylation levels, we found that the UV-induced H3(S10) phosphorylation is almost completely suppressed by inhibition of either MSK1, which can catalyse H3(S10) phosphorylation, or its upstream regulator p38 (Supplementary figure 9e (Immunofluorescence) and Supplementary figure 10c (Immunoblotting)). Therefore, it appears that this is the primary H3(S10) phosphorylation mechanism within this context.

5. The usage of different enzymes/kinase inhibitors should be examined to evaluate the inhibitory efficiency using specific target as readout.

We agree that it is important to show efficiency of inhibition, therefore we had included experiments in the original manuscript to evaluate the efficiency of all inhibitors used in this study. In the original manuscript, inhibitory effects of chromatin modifiers were shown in Supplementary figure 2b (changes in levels of Histone H3 modifications); - of ATM in Supplementary figure 10a (reduced autophosphorylation) and 8b (reduced phosphorylation of ATM's downstream target H2A.X); - and of the p38/MSK1 axis in Supplementary figure 9d (reduced MSK-1 catalysed Histone H3(S10) phosphorylation).

In the revised manuscript, we additionally include Immunoblots showing the efficiency of the ATM, p38 and MSK1 inhibitors (Supplementary figures 8c, 10c).

Reviewer #3:

“In the manuscript titled “Cross talk between chromatin state and ATM signaling in DNA-damage induced transcription stress,” the authors demonstrate how transcription blocking lesion (TBL) activate a series of events, including spliceosome eviction, RNA: DNA hybrid formation, ATM activation, and chromatin modifications including H2AX and H3S10 phosphorylation in non-replicating human dermal fibroblast. Using FRAP and live cell imaging, the authors show that histone acetyl transferases play an important role in mature spliceosome eviction at transcription blocking lesions. Inhibition of histone acetyl transferases leads to suppression of ATM dependent R-loop formation. Moreover, author has shown that ATM activation impact chromatin modification directly by H2AX phosphorylation or indirectly H3S10 phosphorylation via p38/MSK1 signaling pathway. Manuscript is well described with detailed methodology. However, they are several concerns in the current versions.”

We would like to thank the reviewer for her/his appreciation of our work and thoughtful feedback and insightful suggestions, which greatly improved our manuscript.

Major issues:

- 1. It is hard to observe the changes in phosphorylation in representative images and this undermines confidence in the imaging quantitation. For example, Fig.3C sicontrol vs siPCAF and siCTRL vs sip300; Fig. 5A sicontrol +UV 1h vs siATM +UV 1h; Supplementary figure.8a; Fig.4a. Since these events are global phenomena, immunoblots would significantly enhance the manuscript.**

In the revised manuscript we show immunoblotting data (as described above in our response to Reviewers #1 and #2) for the vast majority of our immunofluorescence-experiments confirming our original observations. Also, higher quality images than those included in the supplied PDF document, will also be available if the manuscript is accepted for publication.

However, we would like to emphasize that we quantified IF signals of hundreds of cells in each experiment in order to provide an unbiased measurement that closely reflects the response of the entire population rather than the few random cells shown in the images. This was particularly important for many of our experiments where relatively small differences were observed.

- 1. The places DAPI panel (Fig. 3b group untreated (-UV) and Untreated (+UV); Fig. 4B +UV (FLV)) does not match the antibody group.**

We apologize for this overlook. This has been now corrected in the revised manuscript. Please note that this panel is now part of figure 9c.

2. Does inhibition of ATM or silencing of ATM or UV treated for 3h impact on basal p38 expression ? In Fig. 5A and extended Fig. 10-b, it is important to include basal p38 expression data in each condition.

This is an excellent point. We have not observed changes in basal p38 expression as shown in the representative immunoblots of the newly added experiment in Supplementary figure 10c of the revised manuscript.

Minor issues:

1. Plotted data points are not visible mainly fig. 2a .

We apologize for this inconvenience. We have now changed the colour scheme in the revised manuscript so that the points are more visible in all the figures.

2. Missing y-axis label fig. 2b .

We apologize for the omission; a y-axis label is now added to the graph.

3. In most cases, author has used THZ1 to demonstrate transcription dependent phenotype except Fig. 4B. Why do they perform this assay using other inhibitors? Author is also advised to perform using THZ .

We performed these experiments prior to our observation that ATM plays a role in H3(S10) phosphorylation, but we absolutely agree with the reviewer. We amended the figure to include the THZ1 Immunofluorescence (Figure 4c) and Immunoblotting (figure 4e) experiments. The original figure is now included as Supplementary figure 9c.

4. Page 8-line no.18-19 stating “UVC-induced histone PTMs in non-replicating HDFs, we observed a significant upregulation of H3S10P (extended fig.1b)”. In cited figure, there is roughly 15-20% difference between groups, and it is devoid of any statistical test. Therefore, it is hard to say significant differences. Please justify this sentence.

This is absolutely correct. We modified the sentence, which was an oversight from our part.

5. Page no. 9 line no. 8, there is wrong citation for figure. Fig.4A should be there instead of Fig.4C.

We provide the corrected figure in the revised manuscript.

6. In Supplementary figure 2 legend, (F) and (G) is wrongly placed.

Once again, our apologies for the oversight. The placement of the legend is now corrected.

7. During transcription blocking events, histone acetylation transferases including p300, pCAF evicts spliceosome and forming the aberrant R-loop formation and ATM signaling events. Extended figure 7, shown that some ATM molecules are phosphorylated independently to R-loop, and relaxed histone induces pATM. Does it possible ATM directly impact on spliceosome eviction by changes on chromatin conformation?

If we understand correctly, there are two levels in this question. First, the source of active (auto-phosphorylated ATM) in unperturbed (not UV-damaged) cells and in response to chromatin perturbation irrespective of DNA damage. Second whether ATM influences spliceosome kinetics by changes in chromatin conformation.

In response to the first part of the question, we have consistently observed background levels of active ATM in unperturbed cells. This is not unexpected, as ATM is known to be involved in maintaining cellular homeostasis and orchestrate the response to endogenous damage, particularly oxidative stress induced by normal metabolic activities (e.g. Tanaka et al, PMID: 16940754, Guo et al PMID: 21150274).

Changes in chromatin state have been proposed to activate ATM since its discovery (Bakkenist and Kastan, PMID: 12556884) although the mechanism was entirely unknown at that time. Here, we show that exogenously induced chromatin relaxation disrupts the coupling of the splicing machinery with the elongation complex. This disruption could lead to aberrant R-loop formation, as intron-retaining nascent RNA, lacking bound proteins, can easily invade the negatively supercoiled dsDNA strand upstream of the polymerase. Our collective data suggest that these 'aberrant' R-loops act as signals for ATM activation.

Regarding the second question, indeed it is highly likely that ATM influences spliceosome mobility by promoting chromatin relaxation. Given the broad, context-dependent functions of ATM, numerous mechanisms could underlie this effect, either at a global level or at specific genomic loci, through both direct and indirect means. For instance, during double-strand break (DSB) repair, ATM-mediated phosphorylation of KAP1 loosens chromatin structure, allowing repair machinery better access to damaged DNA. More relevant to the findings in this manuscript, while the phosphorylation of histone variant H2A.X or histone H3(S10) does not directly affect chromatin compaction, both modifications have been shown to do so indirectly. In response to DSBs, ATM-

phosphorylated H2A.X serves as a “platform” for the recruitment of chromatin remodelers and repair factors. Similarly, during the transcriptional activation of stress-response genes, phosphorylation of H3(S10) has been implicated in promoting histone acetylation and subsequent chromatin relaxation. In the revised manuscript, we amended the “Discussion” to indicate this possibility (page 12, lines 14-16 in the revised manuscript).

Point-by-Point Response to Reviewers

We appreciate the time and consideration the reviewers have dedicated to our work. Below we address the reviewers' final remarks and summarize how the revised manuscript incorporates their feedback.

Reviewer #1 (Remarks to the Author):

"The authors have addressed some of my initial concerns (they have provided many required WB and chromatin/WB controls, despite I am not fully convinced from the CSB associated nucleosome data in 1c and the DRIP in 4g)

However other part of the manuscript are not significantly improved in my opinion (e.g. clarification of different HATs role and mechanism, chromatin relaxation at sites of TBLs, etc.)

Furthermore, I remain of the opinion that, at this stage and in this form, the manuscript is not of significant novelty and clarity for publication as the journal currently stands.

I hope these comments might help authors to further solidify this work for submission to this or other journals."

We appreciate the reviewer's perspective; however, we had carefully performed all the recommended experiments and incorporated all the requested revisions, as detailed in our rebuttal and the updated manuscript. To further clarify potentially ambiguous points, we have expanded the relevant descriptions and modified the presentation of our findings in the revised version to include two additional main figures (previously supplemental), *Fig 3* and *Fig 4*.

"Clarification of HATs' role and mechanism"

We respectfully disagree with the assertion that the roles and mechanisms of different HATs were insufficiently addressed. Our data clearly demonstrate that p300 is the only histone acetyltransferase whose co-localization with chromatin and UV-induced DNA damage is fully dependent on active transcription (see original Supplementary Figs 4B, 4C; now presented in revised *Figs 4A, 4B*). This transcription dependency strongly supports a specific role for p300

in the cellular response to transcription-blocking lesions (TBLs), as opposed to the transcription-independent localization observed for PCAF and GCN5 at photolesions. We have revised the manuscript to emphasize these distinctions more clearly and have moved the key supporting data into the main figures of the revised manuscript (*Figs 3 and 4*).

“Chromatin relaxation at sites of TBLs”

We acknowledge that we do not provide a direct physical measurement of chromatin relaxation. Instead, our data rely on well-characterized histone acetylation marks, which are established indicators of relaxed chromatin. As previously discussed, direct assessment of chromatin compaction at specific TBL sites is technically not feasible due to the stochastic and genome-wide distribution of such lesions. Accordingly, we have replaced the term "chromatin relaxation" with "chromatin acetylation" where applicable, to more precisely reflect the nature of the assays used and the biological endpoint assessed.

Nonetheless, we would like to underscore that:

- (1) CSB-associated mononucleosomes upstream of damage-stalled transcription elongation complexes are enriched in histone acetylation marks associated with relaxed chromatin (see original Fig. 1C; revised *Fig. 1C*), at time points when global transcription is strongly repressed (see Supplementary Fig. 1C; revised *Fig. EV1C*).
- (2) Disrupting the deposition of these relaxed chromatin-associated histone modifications—either genetically or pharmacologically—interferes with key downstream events of the TBL-specific DNA damage response (DDR), including spliceosome remodeling, R-loop formation, and ATM activation.

“Novelty and clarity”

Regarding the manuscript’s novelty and clarity, we respectfully maintain that the study presents a distinct conceptual advance. Specifically, it demonstrates how chromatin modifications both initiate and sustain the response to transcription-blocking lesions—an area that remains underexplored in the literature, particularly regarding their impact on spliceosome dynamics and ATM signaling. This important point is further emphasized in the current manuscript which was also edited to more clearly articulate the interpretation and significance of our findings. Given the broad interest in transcription stress, chromatin

biology, and DDR signaling, we believe these findings represent both a novel and timely contribution to the field.

Reviewer #2 (Remarks to the Author):

“In the revised manuscript, authors have included a series of cell fractionation and immunoprecipitation experiments combined with immunoblotting (see also response to reviewer #1) to confirm the microscopy-based evidence. Additionally, authors emphasized the reliability and rationality of largely used microscope-based staining in this manuscript by citing their previous Nature paper.

Although the reason seems to be acceptable, it is still necessary to expand the significant of this study. As the direct relationship between ATM signaling and phosphorylation of H3(S10) is the main finding of this manuscript, clearly demonstrating the regulation of ATM to MSK1-phosphorylated H3(S10) is not “beyond the scope of the current study”.

We thank the reviewer for acknowledging our efforts to substantiate the microscopy-based findings with complementary biochemical assays.

While the reviewer highlights the ATM–MSK1–H3S10ph connection as a key feature, we wish to clarify that this represents only one part of a broader signaling pathway. The central finding of the manuscript is not solely the link between ATM and H3S10 phosphorylation, but rather the identification of chromatin modification as both an initiator and effector of TBL-specific DDR signaling. Our data establish how transcription blocking DNA damage induced chromatin acetylation promotes spliceosome eviction, R-loop formation, and ATM activation—integrating upstream transcription stress with downstream chromatin remodeling.

With regard to ATM–MSK1–H3S10 signaling, we have included additional explanatory text in the revised manuscript clarifying our mechanistic interpretation. Specifically, our data show that:

(1) MSK1-catalyzed H3S10 phosphorylation is completely suppressed by inhibition of p38MAPK, its known upstream activator in this context (Supplementary Figs 9E, 9C; revised Figs EV3E, EV5A).

(2) Inhibition of either p38MAPK or ATM blocks activating phosphorylation of MSK1 (original Fig. 5B; revised Fig 7B, and Supplementary Figs 10B and 10C; revised Figs EV5A, EV5C).

(3) ATM inhibition prevents dual phosphorylation of p38MAPK (Thr180/Tyr182), necessary for its full activation (original Fig. 5A; revised Fig. 7A and original Supplementary Figs 10A, 10C; revised Figs EV5A, EV5C).

As ATM is not a dual-specificity kinase, we interpret this to mean that ATM does not act directly on p38, but rather upstream, most likely through activation of MAP3Ks such as TAOs, which have been shown to be ATM substrates and p38 regulators (Raman et al., EMBO J 2007, DOI: 10.1038/sj.emboj.7601668).

This interpretation is now explicitly discussed in the manuscript to clarify our reasoning and acknowledge this signaling link more fully.

Reviewer #3 (Remarks to the Author):

“The authors have addressed my concerns, and I have no further issues with this manuscript.”

We thank the reviewer for the positive evaluation and appreciate the constructive feedback throughout the revision process.

Dr. Maria Tresini
Erasmus MC
Department of Molecular Genetics
Dr Molewaterplein 50
Rotterdam 3015 CN
Netherlands

5th May 2025

Re: EMBOJ-2025-120849-T
Crosstalk between chromatin state and ATM signalling in DNA damage-induced transcription stress

Dear Dr. Tresini, dear Wim,

Thank you again for transferring your previously reviewed and revised manuscript to The EMBO Journal. I have carefully read the study, your response to the original referee reports, as well as the second round of comments from the previous referees; and I have now also had the chance to discuss all of this with my colleagues. We appreciate the key technical issues raised during the initial round of review appear to have been adequately addressed, and that they did no longer draw major referee criticisms. Although two of the referees appear to insist on the addition of deeper mechanistic follow-up work, we feel that this would fall beyond the scope of the present manuscript and not justify further delay of publication of the key new findings of this study at this point. I am therefore happy to say that we would be happy to accept this work for The EMBO Journal, following a number of editorial modifications as detailed below:

Scientific points:

- Please provide a brief point-by-point response answering to the final round of comments from the previous journal; and consider whether any textual modifications may still be warranted in response.

Editorial points (please, also refer to the Guidelines at the end of this email, and our online Guide to Authors, as sticking to them should greatly facilitate editorial processing at the time of resubmission):

- Please download (see link below) our author checklist, and upload it in completed form with the final manuscript.

- Please upload the manuscript text (including figure legends) as an editable text file, and all figures without legends as individual image files with sufficient resolution/quality for production.

- Please adjust the order of the manuscript sections: Title page with complete author information, Abstract, Keywords, Introduction, Results, Discussion, Methods, Data Availability, Acknowledgements, Disclosure and Competing Interests Statement, References, Main Figure Legends, Tables, Expanded Figure Legends.

- Please adjust the format of the reference list and of the in-text citations according to EMBO Journal format (alphabetical order, author name et al + year...); and make sure that all references are complete with year, volume, and page numbers.

- Please note that Materials and Methods need to be described in the main text using our 'Structured Methods' format (for detail, see <https://www.embopress.org/page/journal/14693178/authorguide#structuredmethods>). The in-text "Methods" section should contain method and protocol descriptions (ideally using a step-by-step protocol format to facilitate adoption of the methodologies across labs), while all key reagents, experimental models, software and relevant equipment - including their sources and relevant identifiers - should be listed in a separately uploaded Reagents and Tools Table, a dedicated template for which can be downloaded from the above-linked section of our Author Guidelines.

- Please refer to our author guide (www.embopress.org/page/journal/14602075/authorguide#expandedview) regarding "supplementary figures", and consider re-organizing the current figures and supplemental figures. We are not limited to 5 main figures (e.g. 7 or 8 would be equally fine); in addition, we can include "Expanded View" figures, whose legends would also need to be in the main text, and which would be type-set and directly visible (expandable) with the HTML version of the paper. Additional information beyond the main manuscript and Expanded View (additional figures with legends, tables, methods...) can be included in a single Appendix PDF, which should be pre-faced by a brief Table of Contents. For information on how to call out these different items in the text, please again refer to the above-linked section of our author guidelines.

- Please include information on author affiliations and correspondence on the title page. Furthermore, as we are switching from a free-text author contribution statement towards a more formal statement based on Contributor Role Taxonomy (CRediT) terms, please remove the present Author Contribution section and instead specify each author's contribution(s) directly in the Author

Information page of our submission system during upload of the final manuscript. See <https://casrai.org/credit/> for more information.

- Please shorten the list of keyword terms to 5, ideally using general/conceptual terms, and avoiding combination of spelt-out and abbreviated versions.
- Please rename the Conflict of Interest section into "Disclosure and Competing Interests Statement", in accordance with our updated Guide to Authors (<https://www.embopress.org/competing-interests>)
- Please double-check to make sure to all relevant funding information in the manuscript is congruent with the info entered into our submission system.
- In the Data Availability, please only refer to datasets generated in the current study; while previously deposited dataset should receive a formal "data citation" as part of the reference list, and in the text/methods/legends as appropriate - please follow the detailed guidelines given at <https://www.embopress.org/page/journal/14602075/authorguide#referencesformat>
For microscopy data, please consider uploading them to a public repository (a separate email regarding Source Data provision and curation, with instructions on how to prepare and upload relevant image and numerical raw data, will follow). In case there should be no data deposition to public repositories linked to the study, please state this as "This study includes no data deposited in external repositories."
- Please provide suggestions for a short 'blurb' text prefacing and summing up the conceptual aspect (background, advance) of the study in two sentences (max. 250 characters), followed by 3-5 one-sentence 'bullet points' with brief factual statements of key results of the paper; they will form the basis of an editor-written 'Synopsis' accompanying the online version of the article. Please also upload a synopsis image, which can be used as a "visual title" for the synopsis section of your paper. The image should be in PNG or JPG format, and please make sure that it remains in the modest dimensions of (exactly) 550 pixels wide and 300-600 pixels high.
- In the figure legends, please make sure to include the exact p-values for figures 1A, E, F, G, H; 2A, B, C, D, E; 3A-D; 4A, C, F; 5A, B.
Also, individual figure panels need to be listed in alphabetical order, which currently is not the case in Figure 2 - please correct. Maybe consider mentioning details applying to several panels in a separate paragraph "Data information: xxx" at the end of the legend of Figure 2. Also keep in mind that all references to figures and individual panels in the main text should appear in a sequential manner.
- Finally, our routine image checks found cases of image re-use that needs to be clearly stated/explained/justified in all respective figure legends. This concerns two of the three "H3" loading controls in Fig S4; and between Figs 3b and S6a, where images may have additionally been rotated/flipped. Higher-resolution files and appropriate Source Data shall be very important to better explain this, and to allow for more reliable checking of all remaining figures.

In this light, I am returning the manuscript to you now, in order to allow you to make these presentational revisions, and to upload all final files via the hyperlink below. Please, do not hesitate to get back to me with any remaining questions that you may have. Thank you again for the opportunity to consider this work for The EMBO Journal, and I look forward to receiving your final version.

With kind regards,

Hartmut

1) Every manuscript requires a Data Availability section (even if only stating that no deposited datasets are included). Primary datasets or computer code produced in the current study have to be deposited in appropriate public repositories prior to resubmission, and reviewer access details provided in case that public access is not yet allowed. Further information: [embopress.org/page/journal/14602075/authorguide#dataavailability](https://www.embopress.org/page/journal/14602075/authorguide#dataavailability)

9) To facilitate reproducibility and cross-laboratory adoption of methodologies, please structure the Materials & Methods section as outlined in our guide to authors, including a completed Reagents and Tools Table that can be downloaded from our author guidelines as well (<https://www.embopress.org/page/journal/14602075/authorguide#structuredmethods>).

10) Digital image enhancement is acceptable practice, as long as it accurately represents the original data and conforms to community standards. If a figure has been subjected to significant electronic manipulation, this must be clearly noted in the figure legend and/or the 'Materials and Methods' section. The editors reserve the right to request original versions of figures and the original images that were used to assemble the figure. Finally, we generally encourage uploading of numerical as well as gel/blot image source data; for details see: embopress.org/page/journal/14602075/authorguide#sourcedata

In the interest of ensuring the conceptual advance provided by the work, we recommend submitting a revision within 3 months (3rd Aug 2025). Please discuss the revision progress ahead of this time with the editor if you require more time to complete the revisions. Use the link below to submit your revision:

Link Not Available